# Rethinking Causal Ranking: A Balanced Perspective on Uplift Model Evaluation

**Minqin Zhu** [1]  **Zexu Sun** [2]  **Ruoxuan Xiong** [3]  **Anpeng Wu** [1]  **Baohong Li** [1]  **Caizhi Tang** [4]  **Jun Zhou** [4]  **Fei Wu** [1]  **Kun Kuang** [1]

## Abstract

Uplift modeling is crucial for identifying individuals likely to respond to a treatment in applications like marketing and customer retention, but evaluating these models is challenging due to the inaccessibility of counterfactual outcomes in real-world settings. In this paper, we identify a fundamental limitation in existing evaluation metrics, such as the uplift and Qini curves, which fail to rank individuals with binary negative outcomes accurately. This can lead to biased evaluations, where biased models receive higher curve values than unbiased ones, resulting in suboptimal model selection. To address this, we propose the Principled Uplift Curve (PUC), a novel evaluation metric that assigns equal curve values of individuals with both positive and negative binary outcomes, offering a more balanced and unbiased assessment. We then derive the Principled Uplift Loss (PUL) function from the PUC and integrate it into a new uplift model, the Principled Treatment and Outcome Network (PTONet), to reduce bias during uplift model training. Experiments on both simulated and real-world datasets demonstrate that the PUC provides less biased evaluations, while PTONet outperforms existing methods. The source code is available at: https://github.com/euzmin/PUC.

## 1. Introduction

Uplift modeling is widely applied in fields such as marketing, advertising, and customer retention, where the goal is to support personalized decision-making by identifying

[1]College of Computer Science and Technology, Zhejiang University, China [2]Gaoling School of Artificial Intelligence, Renmin University of China, China [3]Department of Quantitative Theory and Methods, Emory University, USA [4]Ant Group, Zhejiang, China. Correspondence to: Kun Kuang <kunkuang@zju.edu.cn>.

*Proceedings of the 42$^{nd}$ International Conference on Machine Learning*, Vancouver, Canada. PMLR 267, 2025. Copyright 2025 by the author(s).

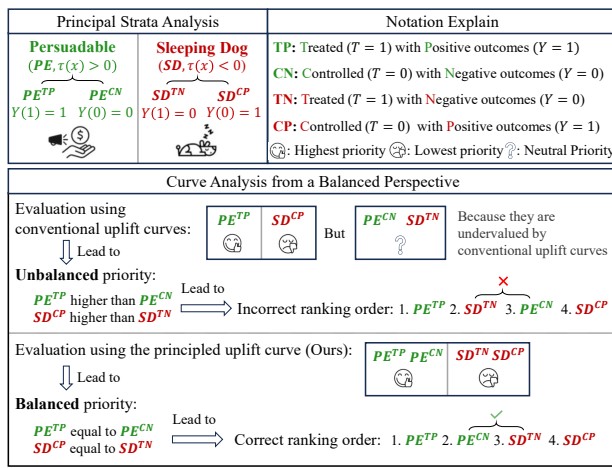

*Figure 1.* A balanced perspective analysis of conventional versus principled curves in evaluating CATE-based principal strata.

individuals most likely to benefit from a treatment (Yamane et al., 2018; Devriendt et al., 2020; Belbahri et al., 2021; Liu et al., 2023). In these scenarios, uplift models estimate the Conditional Average Treatment Effect (CATE) of a binary treatment variable $T$ on a binary outcome variable $Y$ for each individual with covariate $X = x$. The CATE $\tau(x)$ represents the difference in expected outcomes for individuals with covariate $X = x$ assigned to the treatment group ($T = 1$) versus the control group ($T = 0$). Based on CATE values, individuals are divided into two principal strata: persuadables ($PE$, where $\tau(x) > 0$) and sleeping dogs ($SD$, where $\tau(x) < 0$). Each stratum can be further divided into two subgroups according to the outcome value, as illustrated in the principal strata analysis in Figure 1. Uplift modeling aims to rank individuals based on CATE, prioritizing persuadables over others, particularly over sleeping dogs. This indicates the core objective of uplift modeling lies in the *ranking* of CATE values rather than their precise estimation.

However, CATE is a counterfactual quantity and cannot be directly observed in real-world data, posing challenges in its accurate ranking evaluation and using it to make informed decisions (Morgan & Winship, 2015; Xiong et al., 2024). Previous studies have proposed various methods to estimate the CATE, commonly using uplift and Qini curves as evalua-

tion metrics to assess model performance (Shalit et al., 2017; Yamane et al., 2018; Künzel et al., 2019; Shi et al., 2019; Ke et al., 2021; Zhang et al., 2021; Zhong et al., 2022; Liu et al., 2023). These curves evaluate uplift models by quantifying the cumulative gain among ranked individuals, but crucially, only individuals with positive outcomes ($Y = 1$) are explicitly included in the calculation of the uplift and Qini curves. Specifically, these individuals are ranked in descending order according to their estimated CATE, and model performance is measured by calculating the cumulative difference between the number of treated ($T = 1$) and control ($T = 0$) individuals within this ranked subset.

In this paper, we argue that conventional uplift and Qini curves fail to accurately measure the CATE ranking performance of uplift models, largely because they overlook individuals with negative outcomes in their calculations. As illustrated in Figure 1, these metrics focus exclusively on individuals with positive outcomes ($PE^{TP}$ and $SD^{CP}$) in the ranked data, while neglecting those with negative outcomes ($PE^{CN}$ and $SD^{TN}$). As a result, the distinction between persuadables and sleeping dogs among individuals with negative outcomes is overlooked, allowing sleeping dogs in the treated group with negative outcomes ($SD^{TN}$) to be assigned equal or even higher priority than persuadables in the control group with negative outcomes ($PE^{CN}$) in the model ranking. This imbalance enables biased uplift models to attain artificially inflated curve values simply by ranking treated individuals with positive outcomes ($TP$) above others, potentially surpassing unbiased models that more accurately differentiate between these subgroups.

To address this limitation, we propose the Principled Uplift Curve (PUC), a novel evaluation metric that assigns equal importance to individuals with both positive and negative outcomes. As shown in Figure 1, the PUC achieves a balanced assessment by assigning equal value to $PE^{TP}$ and $PE^{CN}$, as well as to $SD^{CP}$ and $SD^{TN}$. This can then provide a more accurate evaluation than conventional uplift and Qini curves. To directly leverage the PUC in model training, we introduce the Principled Uplift Loss (PUL) function and incorporate it into our uplift modeling framework, the Principled Treatment and Outcome Network (PTONet). Specifically, PTONet employs a three-headed neural network as its backbone and incorporates the targeted regularizer from Shi et al. (2019) alongside our proposed principled uplift loss function to enhance the model's ability to rank CATEs effectively. We conduct extensive experiments on simulated data, real-world Criteo data, and real-world Lazada data to demonstrate the superior performance of the Principled Uplift Curve and PTONet.

**Related Work.** Recent research in uplift modeling includes meta-learners (Künzel et al., 2019), tree-based methods (Rzepakowski & Jaroszewicz, 2010; 2012), and deep learn-

ing methods (Shalit et al., 2017; Shi et al., 2019; Devriendt et al., 2020; Ke et al., 2021; Zhong et al., 2022; Liu et al., 2023). A notable study by Renaudin & Martin (2021) cautions against using conventional uplift and Qini curves for evaluating observational data, but suggests applying them in randomized controlled trials (RCT). However, we demonstrate that these curves can still produce biased evaluations even with RCT data. To address this limitation, we propose a novel evaluation metric to mitigate this concern. The most relevant to ours is Devriendt et al. (2020), which designs an uplift helper function to guide the uplift modeling. Building on this, we propose novel loss functions to demonstrate how uplift models optimized to maximize conventional curves can lead to biased estimates. In contrast, our method eliminates these biases, providing a more accurate ranking of CATE. A more detailed discussion of related work is provided in Appendix A.

## 2. Preliminaries

### 2.1. Problem Setup

In uplift modeling, particularly in the context of randomized controlled trials (RCTs), we observe $n$ units in the dataset $D = \{(x_i, t_i, y_i)\}_{i=1}^n$ where $x_i \in \mathcal{X} \subset \mathbb{R}^q$ represents baseline covariates (e.g., product features), $t_i \in \mathcal{T} = \{0, 1\}$ denotes binary treatment assignment (e.g., whether an advertisement is delivered), and $y_i \in \mathcal{Y} = \{0, 1\}$ indicates the binary outcome (e.g., whether the product is purchased). Here, $q$ is the dimension of covariates.

Following the Neyman-Rubin potential outcome framework (Rubin, 1974; Rosenbaum & Rubin, 1983), we denote $Y_i(t)$ as the outcome for unit $i$ under treatment $T = t$. The individual treatment effect (ITE) is defined as $\tau_i = Y_i(1) - Y_i(0)$. Since only one of the potential outcomes, $Y_i(1)$ and $Y_i(0)$, can be observed, the ITE is unobservable. Instead, uplift models estimate the Conditional Average Treatment Effect (CATE), defined as: $\tau(x) = \mathbb{E}[Y_i(1) - Y_i(0) \mid X_i = x]$. We adopt standard causal assumptions (Imbens & Rubin, 2015) to ensure the identifiability of the CATE. Based on the CATE value, individuals can be classified into four strata:

- Persuadable ($\tau(x) > 0$) : $Y(1) = 1$ and $Y(0) = 0$;

- Sure thing ($\tau(x) = 0$): $Y(1) = 1$ and $Y(0) = 1$;

- Lost cause ($\tau(x) = 0$) : $Y(1) = 0$ and $Y(0) = 0$;

- Sleeping dog ($\tau(x) < 0$) : $Y(0) = 1$ and $Y(1) = 0$.

We denote the persuadable, sure thing, lost cause, and sleeping dog strata as $PE$, $ST$, $LC$, and $SD$, respectively. As shown in Figure 1, uplift modeling primarily focuses on the $PE$ and $SD$ strata, as these groups have non-zero CATE values. Superscripts $TP$, $TN$, $CP$, and $CN$ denote treatment ($T = 1$) and control ($T = 0$) groups with positive ($Y = 1$)

or negative ($Y = 0$) outcomes. For example, $PE^{TP}$ represents the persuadable individuals in the treatment group ($T = 1$) with positive outcomes ($Y = 1$).

After estimating the CATE for all individuals, we rank them in descending order. An unbiased uplift model should rank persuadable individuals above others, especially ahead of sleeping dogs. Optimizing and evaluating this CATE ranking is the primary objective of uplift modeling.

### 2.2. Definitions of Uplift and Qini Curves

Since the true CATE values are unknown in real-world datasets, the evaluation of uplift models depends on user-provided ranking rules. These rules can be formulated by CATE estimates, hard-coded heuristics, or other methods (Yadlowsky et al., 2024). We define the ranking rule as:

**Definition 2.1.** A ranking rule is defined by a score function $S : \mathcal{X} \times \mathcal{T} \times \mathcal{Y} \to \mathbb{R}$, where samples $i = 1, \ldots, n$ are ranked in descending order of $S(D_i)$. A higher value of $S(D_i)$ indicates higher priority for treatment. Let $i(j)$ denote the sample corresponding to rank $j$, and $\pi(k, S) = \{i(j)\}_{j=1}^{k}$ represent the top $k$ individuals based on ranking score $S$.

The uplift and Qini curves are commonly used to evaluate the individual rankings $\pi$ of uplift models. During evaluation, three key ranking rules are considered:

- Model ranking rule: $S_{\text{Model}}(D_i) = \hat{\tau}(x_i)$, based on the predicted CATE from the uplift model $\hat{\tau}(x_i)$;

- Random ranking rule: $S_{\text{Random}}(D_i) = \text{Random}(D_i)$, obtained by randomly permuting the dataset $D$;

- Max ranking rule: $S_{\text{Max}}(D_i) = \mathbb{I}(y_i = 1)(\mathbb{I}(t_i = 1) - \mathbb{I}(t_i = 0))$, which assigns highest priority to individuals who experienced a positive outcome under treatment and penalizes those with positive outcomes under control.

Given the ranking rules, we define distinct value indices of the uplift and Qini curves as follows:

$$I = \{i \mid S(D_{i+1}) \neq S(D_i), i \in \pi(n-1, S)\} \cup \{0, n\}. \quad (1)$$

The inclusion of $\{0, n\}$ ensures full coverage of the curve's endpoints. The set $I$ identifies the positions where the uplift score changes (breakpoints), enabling efficient computation of the curve's cumulative values. We denote $I(k)$ as the index of the $k$-th district value.

Following Devriendt et al. (2020), we define the cumulative number of individuals with positive outcomes in treatment and control groups based on the ranking rule $S$ as follows:

$$R_S^T(D, k) = \sum_{i=I(1)}^{I(k)} \mathbb{I}(t_i = 1)\mathbb{I}(y_i = 1),$$
$$R_S^C(D, k) = \sum_{i=I(1)}^{I(k)} \mathbb{I}(t_i = 0)\mathbb{I}(y_i = 1). \quad (2)$$

Based on this definition, the value function for the commonly used Separate Uplift Curve (SUC) (Diemert Eustache et al., 2018) is defined as:

$$V_{\text{SUC}}(k, S) = \frac{R_S^T(D, k)}{|T|} - \frac{R_S^C(D, k)}{|C|}, \quad (3)$$

where the notations $|T|$ and $|C|$ represent the total number of individuals in the treatment and control groups, respectively.

Without loss of generality, we use SUC as the representative of the conventional curve. Other commonly used metrics include the Separate Qini Curve (SQC) (Radcliffe, 2007; Radcliffe & Surry, 2011), Joint Uplift Curve (JUC) (Gutierrez & Gérardy, 2017), and Joint Qini Curve (JQC) (Devriendt et al., 2020), The specific formulations of these metrics are provided in Appendix B.

To quantify the performance of uplift models, we further define the Area metric as follows:

$$Area(S) = \sum_{k=1}^{|I|-1} \frac{(I(k+1) - I(k))}{2}(V(k, S) + V(k+1, S)), \quad (4)$$

where $|I|$ is the total number of indexes in $I$. This Area metric represents the integral of the curve $V(k, S)$ over the intervals $I(k)$. Then, we employ the Area Under the Uplift or Qini Curve (AUUQC) (Gutierrez & Gérardy, 2017; Zhang et al., 2021), defined as follows:

$$\text{AUUQC}(S) = \frac{Area(S_{\text{Model}}) - Area(S_{\text{Random}})}{Area(S_{\text{Max}}) - Area(S_{\text{Random}})}. \quad (5)$$

AUUQC represents the ratio of the area enclosed by the curves of the Model ranking rule and the Random ranking rule to the area enclosed by the curves of the Max ranking rule and the Random ranking rule.

## 3. Can Uplift and Qini Curves Truly Assess Causal Effect Rankings?

Conventional uplift and Qini curves are commonly applied in uplift model evaluation because they approximate the true Average Treatment Effect (ATE) when the CATE ranking is accurate (Yadlowsky et al., 2024). In the absence of true ATE, the magnitude of these curves is commonly used as a proxy for the true ATE, with higher curve values interpreted as indicative of better CATE ranking performance.

Despite their widespread use, a key question arises: Do higher values on conventional uplift and Qini curves truly reflect more accurate CATE rankings? To explore this, we employ simulated data with known individual treatment effects to compare rankings produced by $S_{\text{Model}}$, $S_{\text{Random}}$, and $S_{\text{Max}}$, evaluating these rankings with conventional curves.

As shown in Figure 2, the worst and best cases emerge when ranking $PE^{CN}$ and $SD^{TN}$ using $S_{\text{Max}}$ from SUC, where these two strata are assigned equal priority and thus appear

*Table 1.* The contribution of different individuals to the value functions of various uplift and Qini curves. Columns $T$ and $C$ represent the total contributions from the treatment and control groups, respectively. The contribution of $PE^{TP}$ exceeds that of $PE^{CN}$, and $SD^{TN}$ contributes more than $SD^{CP}$. Additionally, the $T$ column has a greater impact than the $C$ column.

| Curve | $PE^{TP}$ | $PE^{CN}$ | $ST^{TP}$ | $ST^{CP}$ | $LC^{TN}$ | $LC^{CN}$ | $SD^{TN}$ | $SD^{CP}$ | $T$ | $C$ |
|---|---|---|---|---|---|---|---|---|---|---|
| SUC | $\frac{1}{|T|}$ | 0 | $\frac{1}{|T|}$ | $-\frac{1}{|C|}$ | 0 | 0 | 0 | $-\frac{1}{|C|}$ | $\frac{2}{|T|}$ | $-\frac{2}{|C|}$ |
| PUC | 1 | 1 | 1 | $-1$ | $-1$ | 1 | -1 | -1 | 0 | 0 |

| Max Curve Evaluation | Individual Rankings of Conventional Curve (index represents the ranking order) |
|---|---|
| Worst ranking case | $1. ST^{TP} \quad 2. PE^{TP} \quad 3. SD^{TN} \quad 5. LC^{CN}$ 
 $4. LC^{TN} \quad 6. PE^{CN} \quad 7. SD^{CP} \quad 8. ST^{CP}$ |
| Best ranking case | $1. PE^{TP} \quad 2. ST^{TP} \quad 3. PE^{CN} \quad 5. LC^{CN}$ 
 $4. LC^{TN} \quad 6. SD^{TN} \quad 7. ST^{CP} \quad 8. SD^{CP}$ |

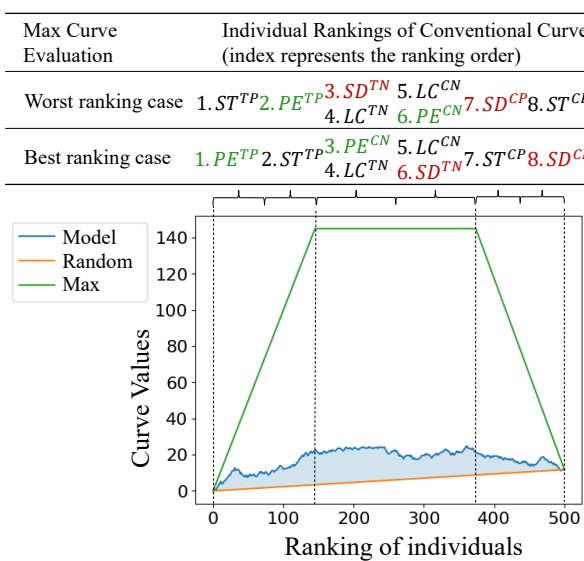

*Figure 2.* The worst and best cases for individual ranking using $S_{\text{Max}}$ in SUC, which arise from the randomness in ranking equally prioritized individuals and reflect the lowest and highest consistency with the true CATE ranking. The axes show individual index and curve values. The blue, yellow, and green lines represent the Model curve (based on $S_{\text{Model}}$), the Random curve (based on $S_{\text{Random}}$), and the Max curve (based on $S_{\text{Max}}$).

in random order. The ranking most consistent with the true CATE, which represents the best case, is achieved when all $PE^{CN}$ individuals are placed ahead of $SD^{TN}$, whereas the least consistent case, representing the worst case, occurs when this order is reversed. Results show that even in the best case, the conventional curves fail to achieve the correct ranking, and in the worst case, they may rank sleeping dogs ahead of persuadable individuals.

To further validate the phenomenon illustrated in Figure 2, we conduct a case study using two uplift models. Table 2 presents the estimated CATE values for eight individuals, labeled $D_1$ through $D_8$, each drawn from a different stratum. Here, $\hat{\tau}^{(1)}$ serves as an unbiased estimator, whereas $\hat{\tau}^{(2)}$ is biased–it captures incorrect causal effects but coincidentally aligns with the ranking rule $S_{\text{Max}}$ of conventional curves. Table 3 displays the AUUQC values of the conventional curves (SUC, SQC, JUC and JQC) and the principled uplift curve (PUC, which will be formally introduced in the next

section). The results show that the AUUQC values for the biased model, $\hat{\tau}^{(2)}$, surpass those of the unbiased model, $\hat{\tau}^{(1)}$, across all conventional curves.[1]

To understand why the conventional curves fail, we quantify the *contribution* of individuals to each curve's value function. For example, in the SUC value function, the contribution of a treatment group individual $i$ with a positive outcome is $\frac{dV_{\text{SUC}}(k,S)}{di} = \frac{1}{|T|} \times 1 - \frac{1}{|C|} \times 0 = \frac{1}{|T|}$, and the total treatment group contribution and control groups are $\frac{2}{|T|}$ and $-\frac{2}{|C|}$. A similar approach applies to SQC, JUC, JQC and PUC.[2] As shown in Table 1, in the SUC row, the

*Table 2.* Case study of two uplift models, $\hat{\tau}^{(1)}$ and $\hat{\tau}^{(2)}$, with their estimated CATE values for eight individuals, labeled $D_1$ to $D_8$, each drawn from different subgroups.

| Uplift Model | $D_1$ $PE^{TP}$ | $D_2$ $PE^{CN}$ | $D_3$ $ST^{TP}$ | $D_4$ $ST^{CP}$ | $D_5$ $LC^{TN}$ | $D_6$ $LC^{CN}$ | $D_7$ $SD^{TN}$ | $D_8$ $SD^{CP}$ |
|---|---|---|---|---|---|---|---|---|
| $\hat{\tau}^{(1)}$ | 1 | 1 | 0 | 0 | 0 | 0 | -1 | -1 |
| $\hat{\tau}^{(2)}$ | 1 | 0 | 1 | -1 | 0 | 0 | 0 | -1 |

*Table 3.* AUUQC values for different curves across eight samples ranked by two uplift models. Bold indicates optimal performance, while underlined values denote suboptimal performance. The symbol (↑) indicates that higher AUUQC values are desirable.

| Uplift Model | SUC (↑) | SQC (↑) | JUC (↑) | JQC (↑) | PUC (↑) |
|---|---|---|---|---|---|
| $\hat{\tau}^{(1)}$ | 0.500 | 0.500 | 0.667 | 0.500 | **0.613** |
| $\hat{\tau}^{(2)}$ | **1.000** | **1.000** | **0.833** | **1.000** | 0.484 |

$TP$ individuals *contribute more* to the conventional curves than others, while $CP$ individuals *contribute least*. This focus on individuals with positive outcomes undervalues the $PE^{CN}$ relative to $PE^{TP}$ while overvaluing $SD^{TN}$ relative to $SD^{CP}$, ultimately making the value of $PE^{CN}$ comparable to that of $SD^{TN}$. This imbalance even leads to a higher overall contribution from the individuals in the treatment group compared to those in the control group. Thus, an uplift model that maximizes conventional curves will rank $TP$ individuals at the top, $CP$ individuals at the bottom, and all other individuals in between. This ranking strategy can lead

---

[1]For detailed calculation process of the conventional and principled curves, please refer to Appendix C.

[2]For the details, please refer to Appendix D.

to incorrect rankings, such as placing $PE^{CN}$ individuals below $SD^{TN}$ individuals.

This leads to an important conclusion: an incorrect ranking can cause the AUUQC of conventional curves to exceed that of a correct ranking. In some cases, it may even produce the maximum AUUC value. Beyond the case studies, this conclusion is further supported by analyzing the conventional curve value functions. For sufficiently large RCT sample sizes, the shared components $\Delta(D, k)$ of conventional curve value functions can be expressed as:

$$\Delta(D,k) = R_S^T(D,k) - R_S^C(D,k)$$
$$= \sum_{i=I(1)}^{I(k)} \mathbb{P}(t_i = 1, y_i = 1) - \mathbb{P}(t_i = 0, y_i = 1). \quad (6)$$

Although this difference in probabilities is related to the true ATE, it is important to note that in the context of CATE ranking, a large value of $\sum_{i=I(1)}^{I(k)} \mathbb{P}(t_i = 1, y_i = 1)$ leads to an increase in the conventional curve value. Consequently, ranking $ST^{TP}$ and $PE^{TP}$ ahead of other individuals creates an incorrect but inflated $\sum_{i=I(1)}^{I(k)} \mathbb{P}(t_i = 1, y_i = 1)$ in the top $I(k)$ individuals, thus exaggerating the conventional curve value. Similarly, ranking all treatment group individuals above control group individuals also inflates the conventional curve value. This inflation effect is further amplified when calculating the AUUQC.

This reveals a fundamental limitation of conventional curves: they are susceptible to manipulation by suboptimal ranking strategies, inflating the performance of biased uplift models. Although the inclusion of denominators such as $|T|$ and $|C|$ in the conventional curve value function partially mitigates this issue, Table 1 shows that it remains insufficient. This explains why the biased model $\hat{\tau}^{(2)}$ shows a higher AUUQC than the unbiased model $\hat{\tau}^{(1)}$ in Tables 2 and 3. In $\hat{\tau}^{(2)}$, the higher ranking of $ST^{TP}$ and $PE^{TP}$ increases the treatment group's contribution, inflating the conventional curve value.

## 4. Principled Uplift Curve and PTONet

Based on the above analysis, we conclude that conventional uplift and Qini curves primarily focus on individuals with positive outcomes, while undervaluing the potential value of individuals with negative outcomes in both the treatment and control groups. The two terms can be formulated as:

$$NR_S^T(D,k) = \sum_{i=I(1)}^{I(k)} \mathbb{I}(t_i = 1)\mathbb{I}(y_i = 0),$$
$$NR_S^C(D,k) = \sum_{i=I(1)}^{I(k)} \mathbb{I}(t_i = 0)\mathbb{I}(y_i = 0). \quad (7)$$

This undervaluation leads to an imbalance in the contributions of individuals from different strata to the curve values, resulting in biased evaluations of existing metrics. For instance, the value of $PE^{TP}$ exceeds that of $PE^{CN}$.

| Max Curve Evaluation | Individual Rankings of Principled Uplift Curve (index represents the ranking order) | | |
|---|---|---|---|
| Worst ranking case | 1. $ST^{TP}$  2. $LC^{CN}$ | 3. $PE^{TP}$  4. $PE^{CN}$ | 5. $SD^{TN}$  6. $SD^{CP}$ | 7. $LC^{TN}$  8. $ST^{CP}$ |
| Best ranking case | 1. $PE^{TP}$  2. $PE^{CN}$ | 3. $ST^{TP}$  4. $LC^{CN}$ | 5. $LC^{TN}$  6. $ST^{CP}$ | 7. $SD^{TN}$  8. $SD^{CP}$ |

*Figure 3.* The worst case and best case for individual ranking of $S_{\text{Max}}$ in PUC. The meaning of the coordinates, legends and table headers is similar to that in Figure 2.

To address this imbalance, we propose the Principled Uplift Curve (PUC), which assigns equal curve values to individuals with both positive and negative outcomes, ensuring a more balanced evaluation. The ranking rule that maximizing PUC value is defined as follows:

$$S_{\text{Max}}(D_i) = \mathbb{I}(y_i = 1)(\mathbb{I}(t_i = 1) - \mathbb{I}(t_i = 0)) + \mathbb{I}(y_i = 0)(\mathbb{I}(t_i = 0) - \mathbb{I}(t_i = 1)). \quad (8)$$

This ranking rule achieves optimality when all samples belonging to the $TP$ and $CN$ groups are ranked ahead of those belonging to the $TN$ and $CP$ groups.

Correspondingly, the value function of the principled uplift curve is formulated as follows[3]:

$$V_{\text{PUC}}(k, S) = R_S^T(D,k) + NR_S^C(D,k) - R_S^C(D,k) - NR_S^T(D,k). \quad (9)$$

The intuition behind this formulation is that persuadable individuals are primarily found in the $TP$ and $CN$ groups, whereas sleeping dogs are mostly in the $TN$ and $CP$. Consequently, $TP$ and $CN$ are ranked above $TN$ and $CP$, with both pairs having equal priority.

The above ranking rule and value function ensure accurate evaluation of uplift models by equitably valuing individuals with positive and negative outcomes.

**Proposition 4.1** (Properties of PUC). *The principled uplift curve exhibits the following properties: 1) It differentiates*

---

[3]We explore the relationship of PUC and conventional curves in Appendix F and detail eight PUC variants in Appendix G.

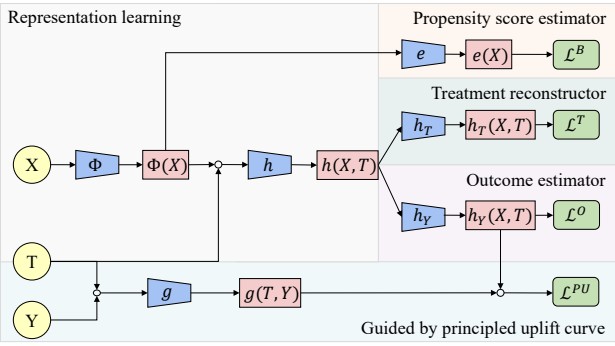

*Figure 4.* The architecture of PTONet. Circles represent random variables; trapezoids denote functions; square rectangles depict function outputs; and rounded rectangles denote losses.

*persuadable individuals from sleeping dogs. 2) Within each of these two strata, individuals from the treatment and control groups contribute equally to the curve value.*

For proof of Proposition 4.1, please see Appendix E.

We can better understand Proposition 4.1 by referring to the earlier examples. As demonstrated in Table 1, although the PUC does not fully distinguish among all four strata, it effectively separates the persuadable and sleeping dog groups by incorporating the value function for individuals with negative outcomes. Furthermore, it ensures consistent value assignment within the treatment and control groups across these strata. As shown in Table 3, the PUC assigns a higher AUUQC score to the unbiased model $\hat{\tau}^{(1)}$ than to the biased model $\hat{\tau}^{(2)}$, demonstrating its ability to more accurately evaluate uplift models. Furthermore, a comparison between Figure 2 and Figure 3 reveals that the PUC consistently ranks persuadable individuals above sleeping dogs, even under the worst-case scenario.

Next, we propose a helper function to guide the uplift modeling. Based on Proposition 4.1, the principled uplift curve suggests that an effective uplift model should satisfy the following principles: 1) rank persuadable individuals above sleeping dogs, and 2) minimize treatment assignment bias.

To address the first principle, inspired by $S_{\text{Max}}$ in the PUC, we design a helper function as follows[4]:

$$g(t_i, y_i) = \mathbb{I}(y_i = t_i), \quad (10)$$

which assigns a value of 1 to samples in the $TP$ and $CN$ groups, and a value of 0 to those in the $TN$ and $CP$ groups.

Using $g(t_i, y_i)$ as a proxy label, we can train a binary classifier to regularize the estimated causal effect $\hat{\tau}(x_i)$ produced

by uplift models. This encourages the model to assign higher values of $\hat{\tau}(x)$ to samples in the $TP$ and $CN$ groups, and lower values to those in the $TN$ and $CP$ groups.

We propose the Principled Uplift Loss (PUL) function to implement this idea, defined as follows:

$$\mathcal{L}^{PU}(D) = \frac{1}{n} \sum_{i=1}^{n} \text{BCE}(g(t_i, y_i), \sigma(x_i)). \quad (11)$$

Here, $\text{BCE}(\cdot, \cdot)$ is a binary cross entropy loss function, and $\sigma(x) = \frac{1}{1+\exp(-\hat{\tau}(x))}$ is the sigmoid function of the estimated CATE $\hat{\tau}(x)$. The PUL function guides the model to assign higher CATE values to persuadable individuals (comprising $PE^{TP}$ and $PE^{CN}$) than to sleeping dog individuals (comprising $SD^{TN}$ and $SD^{CP}$), thereby ranking persuadable individuals ahead of sleeping dogs.

To address the second principle, we design the treatment and outcome loss function $\mathcal{L}^{TO}$, which simultaneously handles treatment and outcome prediction as well as treatment assignment bias. It is formulated as follows:

$$\mathcal{L}^{TO}(D_i) = \mathcal{L}^{T}(x_i, t_i) + \mathcal{L}^{O}(x_i, t_i, y_i) + \mathcal{L}^{B}(x_i, t_i) + \alpha \mathcal{L}^{TR}(x_i, t_i, y_i). \quad (12)$$

The treatment loss function, $\mathcal{L}^{T}(x_i, t_i) = \text{BCE}(t_i, h_T(x_i, t_i))$, aims to reconstruct the treatment assignment, preserving treatment information even in high-dimensional covariate spaces. The outcome loss function, $\mathcal{L}^{O}(x_i, t_i, y_i) = \text{BCE}(y_i, h_Y(x_i, t_i))$, models the outcome given treatment and covariates. The balance loss function, $\mathcal{L}^{B}(x_i, t_i) = \text{BCE}(t_i, e(x_i))$, estimates the propensity score. Finally, the targeted regularizer is defined as $\mathcal{L}^{TR}(x_i, t_i, y_i) = \text{BCE}(y_i, h_Y(x_i, t_i) + \epsilon(\frac{t_i}{e(x_i)} - \frac{1-t_i}{1-e(x_i)}))$, which addresses treatment assignment bias and enhances model scalability.[5] Here, $\epsilon$ is a learnable parameter, and $\alpha$ is a hyperparameter.

Building upon the loss $\mathcal{L}^{TO}$, we design a three-headed learner named Principled Treatment and Outcome Network (PTONet). The architecture of PTONet is presented in Figure 4. The final loss of the PTONet is defined as:

$$\mathcal{L}^{PTO}(D) = \mathcal{L}^{TO}(D) + \beta \mathcal{L}^{PU}(D), \quad (13)$$

where $\beta$ is a hyperparameter.

We emphasize that the core contribution of PTONet lies in the introduction of the PUL function. PTONet is specifically designed as an illustrative uplift model tailored to the PUL function $\mathcal{L}^{PU}$, with its backbone based on a variant of DragonNet (Shi et al., 2019). In particular, we adapt DragonNet by replacing its original two outcome heads with a single head for treatment reconstruction and another for outcome prediction. Detailed descriptions of each module in PTONet are provided in Table 6 and Appendices H and I.

---

[4]To adapt the design of the BCE loss, we set $g(t_i, y_i) = 0$ instead of $g(t_i, y_i) = -1$ for $TN$ and $CP$ individuals.

[5]For a detailed discussion of treatment assignment bias and targeted regularizer, please refer to Appendix H and Section 3 in Shi et al. (2019).

*Table 4.* Performance comparison (mean$_{\pm\text{std}}$) on synthetic data. SUC, SQC, JUC, JQC, and PUC denote their AUUQC values. The symbol (↓) indicates lower is better; (↑) indicates higher is better. The "TRUE" row represents the results of the true CATE. Compared to conventional curves, PUC values more closely approximate AUTGC. Moreover, models guided by PUC, including S-Learner (PU) and PTONet, consistently demonstrate superior performance on both PUC and AUTGC.

| Uplift Model | Unbalanced Conventional Evaluation Metric | | | | | Balanced Evaluation Metric | |
| --- | --- | --- | --- | --- | --- | --- | --- |
| | PEHE (↓) | SUC (↑) | SQC (↑) | JUC (↑) | JQC (↑) | PUC (↑) | AUTGC (↑) |
| TRUE | 0.000 | 0.835 | 0.586 | 0.779 | 0.581 | 1.000 | 1.000 |
| T-Learner | $1.230_{\pm0.17}$ | $0.502_{\pm0.31}$ | $0.352_{\pm0.22}$ | $0.506_{\pm0.29}$ | $0.356_{\pm0.21}$ | $0.614_{\pm0.37}$ | $0.704_{\pm0.28}$ |
| TARNet | $1.293_{\pm0.18}$ | $0.343_{\pm0.30}$ | $0.241_{\pm0.21}$ | $0.344_{\pm0.30}$ | $0.243_{\pm0.21}$ | $0.431_{\pm0.35}$ | $0.565_{\pm0.27}$ |
| CFRNet | $1.246_{\pm0.30}$ | $0.320_{\pm0.38}$ | $0.224_{\pm0.27}$ | $0.301_{\pm0.36}$ | $0.222_{\pm0.26}$ | $0.384_{\pm0.45}$ | $0.528_{\pm0.35}$ |
| DragonNet | $1.055_{\pm0.85}$ | $0.570_{\pm0.25}$ | $0.519_{\pm0.18}$ | $0.570_{\pm0.25}$ | $0.519_{\pm0.18}$ | $0.636_{\pm0.30}$ | $0.739_{\pm0.23}$ |
| EUEN | $1.117_{\pm0.04}$ | $0.407_{\pm0.35}$ | $0.285_{\pm0.24}$ | $0.381_{\pm0.36}$ | $0.269_{\pm0.25}$ | $0.472_{\pm0.43}$ | $0.603_{\pm0.33}$ |
| DESCN | $1.251_{\pm0.23}$ | $0.491_{\pm0.44}$ | $0.404_{\pm0.31}$ | $0.494_{\pm0.43}$ | $0.404_{\pm0.31}$ | $0.418_{\pm0.54}$ | $0.569_{\pm0.41}$ |
| EFIN | $1.868_{\pm0.28}$ | $0.463_{\pm0.29}$ | $0.414_{\pm0.20}$ | $0.428_{\pm0.27}$ | $0.402_{\pm0.20}$ | $0.448_{\pm0.34}$ | $0.648_{\pm0.26}$ |
| S-Learner | $1.209_{\pm0.25}$ | $0.495_{\pm0.36}$ | $0.347_{\pm0.25}$ | $0.476_{\pm0.34}$ | $0.345_{\pm0.25}$ | $0.609_{\pm0.42}$ | $0.700_{\pm0.32}$ |
| S-Learner (U) | $1.009_{\pm0.25}$ | $0.667_{\pm0.31}$ | $0.468_{\pm0.22}$ | $0.631_{\pm0.29}$ | $0.467_{\pm0.21}$ | $0.800_{\pm0.37}$ | $0.847_{\pm0.28}$ |
| S-Learner (PS) | $1.019_{\pm0.23}$ | $0.690_{\pm0.29}$ | $0.484_{\pm0.20}$ | $0.654_{\pm0.27}$ | $0.484_{\pm0.20}$ | $0.828_{\pm0.34}$ | $0.868_{\pm0.26}$ |
| S-Learner (PU) | $\mathbf{0.879_{\pm0.16}}$ | $\mathbf{0.786_{\pm0.16}}$ | $\mathbf{0.552_{\pm0.11}}$ | $\underline{0.738_{\pm0.15}}$ | $\mathbf{0.548_{\pm0.11}}$ | $\underline{0.943_{\pm0.19}}$ | $\underline{0.957_{\pm0.15}}$ |
| PTONet | $\underline{0.883_{\pm0.13}}$ | $\underline{0.780_{\pm0.14}}$ | $\underline{0.547_{\pm0.10}}$ | $\mathbf{0.746_{\pm0.13}}$ | $\underline{0.546_{\pm0.10}}$ | $\mathbf{0.948_{\pm0.15}}$ | $\mathbf{0.961_{\pm0.11}}$ |

# 5. Experiments

We conduct experiments to address the following three key research questions:

- RQ1) Does the principled uplift curve yield results closer to the true CATE than other metrics?

- RQ2) Does the principled uplift loss function effectively improve the CATE ranking ability of uplift models?

- RQ3) How does the performance of PTONet compare to existing methods?

## 5.1. Experimental Setup

We compare the performance of uplift models and their evaluation curves using a synthetic dataset and a real-world Criteo dataset (Diemert Eustache et al., 2018; Diemert et al., 2021). In Appendix I, we present experimental results on the Lazada dataset (Zhong et al., 2022) to validate the scalability of the proposed method on higher-dimensional data. For all datasets, we conducted 50 performance comparison experiments, changing the random seed from 0 to 49.

**Dataset Description.** The synthetic dataset contains $n = 10,000$ units with $q = 10$ different covariates. The real-world Criteo dataset (Diemert Eustache et al., 2018; Diemert et al., 2021), open sourced by Criteo AI Labs, is utilized for uplift modeling in a large-scale advertising scenario. It includes 25,309,483 instances with 11 continuous features, a binary treatment, and 2 candidate outcomes (visits and conversions). For our analysis, we specifically consider the visits variable as the outcome. We split the two datasets into

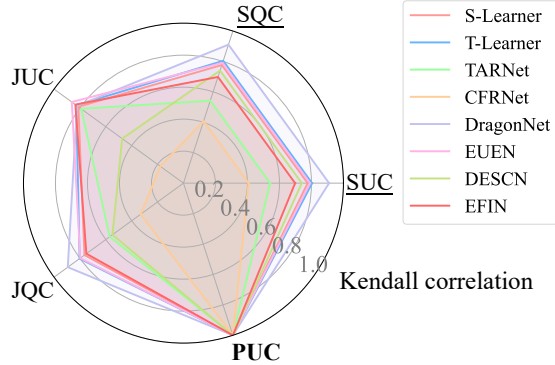

*Figure 5.* The Kendall (higher values are desirable) between AUUQC and AUTGC sequences of uplift models on synthetic data, evaluated for SUC, SQC, JUC, JQC, and PUC. PUC achieves the highest Kendall, followed by SUC and JUC.

training, validation, and test sets in an 8/1/1 ratio.

**Evaluation Metrics.** In addition to the uplift and Qini curves discussed in Table 7, we introduce three metrics for the model evaluation and curve evaluation.

The first is the expected Precision in Estimation of Heterogeneous Effect (PEHE, (Shalit et al., 2017)), defined as:

$$\text{PEHE} = \mathbb{E}[(\hat{\tau}(X) - \tau(X))^2]. \qquad (14)$$

This metric does not account for prioritization, making it an unbalanced conventional evaluation metric.

The second metric is the Area Under True Gain Curve

*Table 5.* Performance comparison (mean$_{\pm\text{std}}$ of AUUQC) on Criteo data. PTONet achieves superior performance on PUC.

| Uplift Model | Unbalanced Conventional Evaluation Metric | | | | Balanced Metric |
|---|---|---|---|---|---|
| | SUC (↑) | SQC (↑) | JUC (↑) | JQC (↑) | PUC (↑) |
| S-Learner | $0.508_{\pm 0.01}$ | $\underline{0.105_{\pm 0.01}}$ | $0.033_{\pm 0.01}$ | $0.085_{\pm 0.01}$ | $0.141_{\pm 0.02}$ |
| T-Learner | $0.356_{\pm 0.32}$ | $0.073_{\pm 0.07}$ | $0.022_{\pm 0.02}$ | $0.057_{\pm 0.06}$ | $0.085_{\pm 0.11}$ |
| TARNet | $0.512_{\pm 0.04}$ | $0.103_{\pm 0.01}$ | $\mathbf{0.036_{\pm 0.01}}$ | $\mathbf{0.091_{\pm 0.01}}$ | $0.104_{\pm 0.05}$ |
| CFRNet | $0.276_{\pm 0.29}$ | $0.057_{\pm 0.06}$ | $\underline{0.034_{\pm 0.02}}$ | $0.044_{\pm 0.05}$ | $0.105_{\pm 0.11}$ |
| DragonNet | $\underline{0.518_{\pm 0.02}}$ | $\mathbf{0.107_{\pm 0.01}}$ | $\underline{0.034_{\pm 0.01}}$ | $\underline{0.087_{\pm 0.01}}$ | $0.137_{\pm 0.05}$ |
| EUEN | $0.509_{\pm 0.01}$ | $\underline{0.105_{\pm 0.01}}$ | $0.033_{\pm 0.01}$ | $0.085_{\pm 0.01}$ | $\underline{0.186_{\pm 0.01}}$ |
| DESCN | $\mathbf{0.556_{\pm 0.06}}$ | $0.09_{\pm 0.04}$ | $0.029_{\pm 0.01}$ | $0.076_{\pm 0.02}$ | $0.072_{\pm 0.05}$ |
| EFIN | $0.504_{\pm 0.05}$ | $0.104_{\pm 0.01}$ | $0.033_{\pm 0.05}$ | $0.084_{\pm 0.01}$ | $0.127_{\pm 0.04}$ |
| PTONet | $0.502_{\pm 0.01}$ | $0.103_{\pm 0.01}$ | $\underline{0.034_{\pm 0.01}}$ | $0.086_{\pm 0.01}$ | $\mathbf{0.191_{\pm 0.01}}$ |

*Table 6.* Ablation studies (mean$_{\pm\text{std}}$) of PTONet modules on synthetic data. All modules enhance PTONet's performance.

| Uplift Model | Unbalanced Conventional Evaluation Metric | | | | | Balanced Evaluation Metric | |
|---|---|---|---|---|---|---|---|
| | PEHE (↓) | SUC (↑) | SQC (↑) | JUC (↑) | JQC (↑) | PUC (↑) | AUTGC (↑) |
| TRUE | 0.000 | 0.835 | 0.586 | 0.779 | 0.581 | 1.000 | 1.000 |
| PTONet | $\underline{0.883_{\pm 0.13}}$ | $\mathbf{0.780_{\pm 0.14}}$ | $\mathbf{0.547_{\pm 0.10}}$ | $\mathbf{0.746_{\pm 0.13}}$ | $\mathbf{0.546_{\pm 0.09}}$ | $\mathbf{0.948_{\pm 0.15}}$ | $\mathbf{0.961_{\pm 0.11}}$ |
| w/o $\mathcal{L}^{PU}$ | $1.138_{\pm 0.36}$ | $0.524_{\pm 0.38}$ | $0.368_{\pm 0.27}$ | $0.532_{\pm 0.34}$ | $0.377_{\pm 0.25}$ | $0.659_{\pm 0.45}$ | $0.739_{\pm 0.35}$ |
| w/o $\mathcal{L}^{TR}$ | $\mathbf{0.846_{\pm 0.15}}$ | $\underline{0.760_{\pm 0.20}}$ | $\underline{0.533_{\pm 0.14}}$ | $\underline{0.737_{\pm 0.17}}$ | $\underline{0.533_{\pm 0.13}}$ | $\underline{0.936_{\pm 0.22}}$ | $\underline{0.951_{\pm 0.17}}$ |
| w/o $\mathcal{L}^{B}$ | $1.059_{\pm 0.34}$ | $0.533_{\pm 0.38}$ | $0.374_{\pm 0.26}$ | $0.545_{\pm 0.36}$ | $0.380_{\pm 0.26}$ | $0.678_{\pm 0.46}$ | $0.753_{\pm 0.35}$ |
| w/o $\mathcal{L}^{T}$ | $1.012_{\pm 0.33}$ | $0.604_{\pm 0.35}$ | $0.424_{\pm 0.25}$ | $0.601_{\pm 0.32}$ | $0.429_{\pm 0.24}$ | $0.756_{\pm 0.41}$ | $0.813_{\pm 0.31}$ |

(AUTGC), which is formulated as follows:

$$\text{AUTGC}(S) = \frac{\text{Area}\left(S_{\text{Model}}\right) - \text{Area}\left(S_{\text{Random}}\right)}{\text{Arca}\left(S_{\text{True}}\right) - \text{Arca}\left(S_{\text{Random}}\right)}, \quad (15)$$

where $S_{\text{True}}(x) = \tau(x)$ represents the true CATE ranking.

To assess the alignment between curves and true CATE rankings, the Kendall correlation (Kendall, 1938) between the AUUQC values of $m$ models and their AUTGC values is calculated as follows:

$$\text{Kendall} = \frac{2}{m(m-1)} \sum_{i=1}^{m} \sum_{j=1}^{i-1} \mathbb{I}(\text{AUTGC}_i \\ - \text{AUTGC}_j) \cdot \mathbb{I}(\text{AUUQC}_i - \text{AUUQC}_j). \quad (16)$$

A higher Kendall value indicates a closer alignment of the uplift and Qini curve with the true CATE ranking.

**Baseline Implementations.** We evaluate eight uplift models as baselines, including S-Learner (Künzel et al., 2019), T-Learner (Künzel et al., 2019), TARNet (Shalit et al., 2017), CFRNet (Shalit et al., 2017), DragonNet (Shi et al., 2019), EUEN (Ke et al., 2021), DESCN (Zhong et al., 2022) and EFIN (Liu et al., 2023). All baselines follow the same parameter tuning process, please see Appendix I for details.

### 5.2. Experimental Results

**Comparing Evaluation Metrics through Simulation.** To address RQ1, we assess uplift and Qini curve effectiveness

by evaluating their correlation with AUTGC.

Since AUTGC is unobservable in real-world settings, we use synthetic data. We compute the AUUQC and AUTGC metrics for uplift models trained on synthetic data over $m = 50$ epochs and quantify the rank correlation between these sequences using the Kendall coefficient.

As shown in Figure 5, all AUUQC sequences positively correlate with AUTGC. However, except for the PUC, other metrics fail to fully align with AUTGC trends (Kendall < 1), indicating that high values of conventional curves do not reliably reflect strong CATE ranking performance. In contrast, the PUC achieves perfect alignment with AUTGC (Kendall = 1), making it a reliable metric for evaluating CATE ranking in uplift models.

In addition to the results shown in Figure 5, all subsequent experiments based on synthetic data (including Tables 4 to 6) further validate the alignment between PUC and AUTGC. Notably, some models that perform well on conventional metrics but fail to achieve comparable results on AUTGC. In contrast, models that perform well on the PUC metric consistently exhibit strong performance on AUTGC as well.

**Comparing Loss Functions through Simulation.** To address RQ2, we first verify that uplift models guided by the PUC outperform conventional curves and other metrics. Following Devriendt et al. (2020), we establish an uplift loss

function $\mathcal{L}^U$ to prioritize individuals in the treatment group with positive outcomes over others, and a propensity score loss function $\mathcal{L}^{PS}$ to guide the CATE for treatment group individuals exceeds that of control group individuals. The loss functions are formulated as follows:

$$\begin{aligned}
\mathcal{L}^U(D_i) &= \mathrm{BCE}(\mathbb{I}(t_i = 1)\mathbb{I}(y_i = 1), \sigma(x_i)), \\
\mathcal{L}^{PS}(D_i) &= \mathrm{BCE}(\mathbb{I}(t_i = 1), \sigma(x_i)).
\end{aligned} \quad (17)$$

For a fair performance comparison, we augment the S-Learner with $\mathcal{L}^U$, $\mathcal{L}^{PS}$, and $\mathcal{L}^{PU}$ as S-Learner (U), S-Learner (PS), and S-Learner (PU).

As shown in Table 4, although the S-Learner (PU) does not outperform the S-Learner (U) in terms of AUUQC for SUC, SQC, and JQC, it achieves the highest scores for both AU-UQC of PUC and AUTGC. Furthermore, when compared to other uplift models, introducing the PUL function solely into the S-Learner allows it to outperform all models except PTONet on PUC and AUTGC. These findings suggest that models guided by the PUC metric produce results that are more robust and better aligned with the ground truth. The experimental results in Table 11 of Appendix I provide further empirical evidence supporting this conclusion.

Additionally, we report the best performance of each model across multiple experiment rounds, as shown in Table 12 in Appendix I. The uplift models optimized with $\mathcal{L}^U$ and $\mathcal{L}^{PS}$ outperform the TRUE row in SUC, SQC, JUC, and JQC but underperform in PUC and AUTGC. This confirms that maximizing conventional curve values can introduce bias, leading to an inflated conventional curve value in a biased model. In contrast, our balanced evaluation metric, PUC, remains unaffected by this issue, even when the model is directly trained with the PUL function $\mathcal{L}^{PU}$.

**Comparing the Performance of Uplift Models.** To address RQ3, we report the performance comparison results on the synthetic dataset in Table 4, the Criteo dataset in Table 5 and the Lazada dataset in Table 13.

The results on synthetic data show that PTONet outperforms other uplift models on PEHE, AUUQC of PUC, and AUTGC. For the real-world Criteo and Lazada dataset, despite the absence of the true CATE, the results indicate that PTONet outperforms others on the AUUQC of PUC.

To investigate the function of each module in PTONet, we conduct ablation studies as summarized in Table 6. The results show that $\mathcal{L}^{PU}$, $\mathcal{L}^T$, and $\mathcal{L}^B$ significantly improve PTONet's performance on both PUC and AUTGC. Since $\mathcal{L}^{TR}$ is specifically designed to address implicit treatment assignment bias, its limited effect on RCT data is expected. Among all components, the PUL objective function remains the most critical factor driving PTONet's effectiveness.

# 6. Conclusion

This paper reveals the limitations of previous uplift and Qini curves in evaluating uplift models, showing that they are vulnerable to manipulation by suboptimal ranking strategies, which can inflate the performance of biased models. We propose the Principled Uplift Curve (PUC), a new metric that balances the value of individuals with both positive and negative outcomes, providing a more accurate evaluation of uplift models. We also propose PTONet, a PUC-guided uplift model that directly optimizes the uplift model by maximizing the PUC value. Extensive experiments demonstrate the effectiveness of the PUC for uplift model evaluation and of the PTONet for uplift model optimization.

# Acknowledgments

This work was supported in part by the National Key Research and Development Program of China (2024YFE0203700), National Natural Science Foundation of China (62376243, 62441605), and "Pioneer" and "Leading Goose" R&D Program of Zhejiang (2025C02037). This work was also supported by Ant Group. All opinions in this paper are those of the authors and do not necessarily reflect the views of the funding agencies.

# Impact Statement

We trust that our work offers a meaningful contribution to the uplift modeling and causal ranking community. In this field, the pivotal yet under-addressed challenge is how to assess and guide the formulation of uplift models in real-world data scenarios. The contribution of our paper is that we fundamentally uncover the issues associated with conventional uplift and Qini curves. In response, we propose a novel and effective metric to address these problems, and furnish a strategy for leveraging this metric to optimize uplift models. This work also has positive implications for the application of causal inference methods in real-world domains. According to our findings, in areas such as healthcare and finance, existing evaluation metrics tend to favor individuals who have received treatment or have a purchase history ($Y = 1$), which may inadvertently disadvantage those without such history ($Y = 0$) and encourage decisions that could harm the sleeping dog group. In contrast, our proposed PUC metric, along with the corresponding PTONet model, better balances the interests of both groups. This promotes fairer decision-making and helps protect the sleeping dog group, thereby supporting no-harm policy implementation.

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

## A. Detailed Related Work Discussion

**Uplift Modeling Methods.** Uplift modeling is widely applied in fields such as marketing, advertising, and customer retention (Xu et al., 2023; Zheng et al., 2024; Zhao et al., 2024a;b; Wu et al., 2024; Wang et al., 2024; Sun et al., 2025; Wu et al., 2025). Uplift modeling methods target persuadable individuals while minimizing unnecessary interventions for others, mainly sleeping dog customers (Gutierrez & Gérardy, 2017; Zhang et al., 2021). Current research in uplift modeling primarily encompasses two distinct approaches: One branch focuses on minimizing the estimation error of causal effects. It includes meta-learners (Künzel et al., 2019), which employ existing machine learning techniques to model individuals' responses; tree-based methods (Radcliffe, 2007; Rzepakowski & Jaroszewicz, 2010; Radcliffe & Surry, 2011; Rzepakowski & Jaroszewicz, 2012), which deploy specialized tree or forest structures, employing a variety of metrics for splitting criteria; and deep methods (Shalit et al., 2017; Shi et al., 2019; Devriendt et al., 2020; Ke et al., 2021; Zhong et al., 2022; Liu et al., 2023), which leverage the strengths of neural networks to incorporate more intricate and adaptable architectures in modeling the response to treatment, aiming for improved accuracy in CATE estimation.

The second branch of approaches seeks to directly maximize the uplift and Qini curve while concurrently regressing on observed outcomes. It includes methods that set the objective function based on maximizing the Qini coefficient directly (Belbahri et al., 2021; Betlei et al., 2021), or employing learning-to-rank techniques with constraints aimed at maximizing the Qini measure (Devriendt et al., 2020). The most relevant to our methodology is Devriendt et al. (2020), which designs an uplift helper function to guide the uplift modeling. Inspired by this work, we design distinct helper functions to demonstrate that helper functions that maximize the uplift and Qini curve can lead to biased estimates. Contrarily, our method avoids such biases, offering a more accurate ranking of CATE.

**Evaluation Metrics for Uplift Model.** In evaluating uplift models on real-world data, the uplift and Qini curves are commonly employed, categorized as separate uplift curve (Rzepakowski & Jaroszewicz, 2010; 2012), separate Qini curve (Radcliffe, 2007; Radcliffe & Surry, 2011), joint uplift curve (Gutierrez & Gérardy, 2017), and joint Qini curve (Radcliffe, 2007; Devriendt et al., 2020). A notable study is Renaudin & Martin (2021), which discourages using uplift/Qini curves for observational data evaluation but suggests their applicability for randomized controlled trials (RCT) data. However, we highlight that conventional uplift/Qini curves may yield biased evaluations even when employing RCT data. The curve most similar to ours is the ROCini presented in Verbeken et al. (2022). However, it lacks theoretical foundation and performs poorly in experiments (please refer to Table 9).

In addition, the recently proposed TOC/AUTOC metric by Yadlowsky et al. (2024) has attracted significant attention in the community, as it may help alleviate some of the known limitations associated with conventional uplift modeling metrics. TOC/AUTOC can be interpreted as an extension of the conventional uplift and Qini curves by introducing a threshold $u$ and applying a logarithmic transformation. Specifically, this metric places greater emphasis on the top-ranked individuals. As illustrated in Figure 2 of Yadlowsky et al. (2024), when only the top 10% of the population is considered, the cumulative gain measured by AUTOC exceeds that of the Qini curve. In other words, this approach further amplifies the inherent imbalance issue in conventional uplift and Qini metrics.

Consequently, we propose a novel evaluation metric to mitigate above concerns.

## B. Definitions of Uplift and Qini Curves

*Table 7.* Definitions of commonly applied uplift and Qini curves for uplift model evaluation and selection. The notations $|T|$ and $|C|$ represent the total number of individuals in the treatment and control groups, respectively.

| Curve Type | Formula (Value Function) |
|---|---|
| Separate Uplift Curve (SUC) | $V_{\mathrm{SUC}}(k, S) = \frac{R_S^T(D,k)}{|T|} - \frac{R_S^C(D,k)}{|C|}$ |
| Separate Qini Curve (SQC) | $V_{\mathrm{SQC}}(k, S) = R_S^T(D, k) - R_S^C(D, k)\frac{|T|}{|C|}$ |
| Joint Uplift Curve (JUC) | $V_{\mathrm{JUC}}(k, S) = \left( \frac{R_S^T(D,k)}{N_S^T(D,k)} - \frac{R_S^C(D,k)}{N_S^C(D,k)} \right) \cdot \left( N_S^T(D, k) + N_S^C(D, k) \right)$ |
| Joint Qini Curve (JQC) | $V_{\mathrm{JQC}}(k, S) = R_S^T(D, k) - R_S^C(D, k)\frac{N_S^T(D,k)}{N_S^C(D,k)}$ |

The uplift and Qini curves are employed to evaluate the CATE rankings of uplift models (Gutierrez & Gérardy, 2017; Zhang et al., 2021). In addition to the metrics defined in the main text, we define the cumulative number of treatment and control

group individuals in the RCT data $D$ as follows:

$$N_S^T(D, k) = \sum_{i=I(1)}^{I(k)} \mathbb{I}(t_i = 1),$$

$$N_S^C(D, k) = \sum_{i=I(1)}^{I(k)} \mathbb{I}(t_i = 0),$$

(18)

where $\mathbb{I}(\cdot)$ is an indicator function.

Based on the above definition, we present the value functions for four commonly used implementations of uplift and Qini curves: Separate Uplift Curve (SUC) (Rzepakowski & Jaroszewicz, 2010; 2012), Separate Qini Curve (SQC) (Radcliffe, 2007; Radcliffe & Surry, 2011), Joint Uplift Curve (JUC) (Gutierrez & Gérardy, 2017), and Joint Qini Curve (JQC) (Devriendt et al., 2020) in Table 7.

These four curves share the same $S_{\text{Max}}$ ranking rule, and therefore, they exhibit the same limitations: they tend to underestimate the value of negative outcome individuals, potentially leading to incorrect evaluations. Although they attempt to address this issue using $|T|$, $|C|$, or $N_S^T$ and $N_S^C$, as shown in Table 8, these problems remain significant.

*Table 8.* The contribution of different individuals to the value functions of various uplift and Qini curves. Columns $T$ and $C$ represent the total contributions from the treatment and control groups, respectively.

| Curve | $PE^{TP}$ | $PE^{CN}$ | $ST^{TP}$ | $ST^{CP}$ | $LC^{TN}$ | $LC^{CN}$ | $SD^{TN}$ | $SD^{CP}$ | $T$ | $C$ |
|-------|-----------|-----------|-----------|-----------|-----------|-----------|-----------|-----------|-----|-----|
| SUC | $\frac{1}{|T|}$ | 0 | $\frac{1}{|T|}$ | $-\frac{1}{|C|}$ | 0 | 0 | 0 | $-\frac{1}{|C|}$ | $\frac{2}{|T|}$ | $-\frac{2}{|C|}$ |
| SQC | 1 | 0 | 1 | $-\frac{|T|}{|C|}$ | 0 | 0 | 0 | $-\frac{|T|}{|C|}$ | 2 | $-2\frac{|T|}{|C|}$ |
| JUC | 1/0 | 1 | 1/0 | 0/-1 | -1 | 1 | -1 | 0/-1 | 0/-1/-2 | 2/1/0 |
| JQC | 3 | 2 | 3 | 0 | 1 | 2 | 1 | 0 | 8 | 4 |
| PUC | 1 | 1 | 1 | $-1$ | $-1$ | 1 | -1 | -1 | 0 | 0 |

## C. AUUQC Value Calculation Process

The specific code of these curves is provided in the utils file of the source code. For instance, the calculation formula for the SUC is as follows:

First, we sort the eight samples in descending order based on the value of $\hat{\tau}^{(1)}$, resulting in the order $\{2, 1, 6, 5, 4, 3, 8, 7\}$. Using the district value index formula in Equation (1), we obtain $I = \{0, 2, 6, 8\}$. The corresponding $R_S^T(D, k)$ and $R_S^C(D, k)$ values are $\{0, 1, 2, 2\}$ and $\{0, 0, -1, -2\}$, respectively. Consequently, the $V_{\text{SUC}}(k, S)$ values are $\{0, 0.25, 0.25, 0\}$. The Model value for this set of valid values, computed using the area function, is given by:

$$\text{Area}(S_{\text{Model}}) = 1.5.$$

(19)

Next, we calculate the Max curve value as $\{0, 0.5, 0.5, 0.5, 0\}$ and the corresponding district value indexes are $\{0, 2, 4, 6, 8\}$, yielding an curve value of 3.

For the Random curve, the curve value is calculated as 0, with valid values ranging from $[0, 0]$ to $[4, ATE = 0]$.

Finally, the SUC is computed as:

$$AUUQC = \frac{1.5 - 0}{3 - 0} = 0.5.$$

(20)

Similarly, we sort the eight samples in descending order based on the value of $\hat{\tau}^{(1)}$, resulting in the order $\{3, 1, 7, 6, 5, 2, 8, 4\}$.

The $V_{\text{SUC}}(k, S)$ values obtained are $\{0, 0.5, 0.5, 0\}$, and the corresponding district value indexes are $\{0, 2, 6, 8\}$. The AUUQC value for this set is 3.

Next, we calculate the Max curve values as $\{0, 0.5, 0.5, 0.5, 0\}$ and the corresponding district value indexes are $\{0, 2, 4, 6, 8\}$, yielding an curve value of 3.

For the Random curve, the curve value is calculated as 0, with valid values ranging from $[0, 0]$ to $[4, ATE = 0]$.

Finally, the AUUQC is computed as:

$$AUUQC = \frac{3 - 0}{3 - 0} = 1. \tag{21}$$

## D. Detailed Derivation of Individual Contributions

Let's compute the derivative of $V_{\mathrm{SUC}}(k, S)$ with respect to an individual $i$ where $T = 1$ and $Y = 1$:

$$\frac{dV_{\mathrm{SUC}}(k, S)}{di} = \frac{d}{di} \left( \frac{R_S^T(D, k)}{|T|} - \frac{R_S^C(D, k)}{|C|} \right). \tag{22}$$

By applying the derivative to both terms:

$$\frac{dV_{\mathrm{SUC}}(k, S)}{di} = \frac{1}{|T|} \frac{d}{di} \left( R_S^T(D, k) \right) - \frac{1}{|C|} \frac{d}{di} \left( R_S^C(D, k) \right). \tag{23}$$

So, the derivative of $R_S^T(D, k)$ is:

$$\frac{d}{di} R_S^T(D, k) = \mathbb{I}(t_i = 1)\mathbb{I}(y_i = 1). \tag{24}$$

Similarly, the derivative of $R_S^C(D, k)$ is:

$$\frac{d}{di} R_S^C(D, k) = \mathbb{I}(t_i = 0)\mathbb{I}(y_i = 1). \tag{25}$$

Substituting the derivatives into the expression for $\frac{dV_{\mathrm{SUC}}(k,S)}{di}$, we get:

$$\frac{dV_{\mathrm{SUC}}(k, S)}{di} = \frac{1}{|T|}\mathbb{I}(t_i = 1)\mathbb{I}(y_i = 1) - \frac{1}{|C|}\mathbb{I}(t_i = 0)\mathbb{I}(y_i = 1). \tag{26}$$

For the specific case where $T = 1$ and $Y = 1$ (treatment group, positive outcome), the indicator functions will be: $\mathbb{I}(t_i = 1) = 1, \mathbb{I}(y_i = 1) = 1, \mathbb{I}(t_i = 0) = 0$ and $\mathbb{I}(y_i = 1) = 1$.

Thus, the derivative simplifies to:

$$\frac{dV_{\mathrm{SUC}}(k, S)}{di} = \frac{1}{|T|} \times 1 - \frac{1}{|C|} \times 0 = \frac{1}{|T|}. \tag{27}$$

So, for $T = 1$ and $Y = 1$, the derivative of the uplift function $V_{\mathrm{SUC}}(k, S)$ with respect to $i$ is:

$$\frac{dV_{\mathrm{SUC}}(k, S)}{di} = \frac{1}{|T|}. \tag{28}$$

This derivative indicates that the inclusion of an individual from the treatment group with a positive outcome contributes $\frac{1}{|T|}$ to the rate of change of the uplift value, which is proportional to the number of treated individuals in the dataset.

A similar approach applies to the SQC. For JUC and JQC, where the value functions depend on $N_S^T(D, k)$ and $N_S^C(D, k)$, we simplify the contribution calculation by considering their relative magnitudes, as outlined by Devriendt et al. (2020).

As shown in Table 8, the $TP$ individuals contribute more to the conventional curves than others, while $CP$ individuals contribute least. This focus on individuals with positive outcomes undervalues the $PE^{CN}$ individuals relative to $PE^{TP}$

individuals while overvaluing $SD^{TN}$ individuals relative to $SD^{CP}$ individuals, ultimately making the value of $PE^{CN}$ individuals comparable to that of $SD^{TN}$ individuals. This imbalance even leads to a higher overall contribution from the individuals in the treatment group compared to those in the control group. Thus, an uplift model that maximizes conventional curves will rank $TP$ individuals at the top, $CP$ individuals at the bottom, and all other individuals in between.[6] This ranking strategy can lead to incorrect rankings, such as placing $PE^{CN}$ below $SD^{TN}$.

## E. Proof of Proposition 1

The theoretical guarantee of the PUC and PUL stems from Proposition 4.1 of the original paper, which states that the CATE for the persuadable group should be greater than the CATE for the sleeping dog group. Specifically, PUL influences the PTONet in two ways: it directly balances the value of the positive outcome ($Y = 1$) and negative outcome ($Y = 0$) groups, and it focuses not only on the accuracy of the CATE effect size estimation but also on the comparison of CATE magnitudes corresponding to different individuals.

Following the distributional treatment effect theorem in Kallus (2022; 2023), we define the persuadable stratum and sleeping dog stratum as $\mathbb{P}(Y(1) = 1, Y(0) = 0|X = x)$ and $\mathbb{P}(Y(1) = 0, Y(0) = 1|X = x)$. According to the properties of joint probability, we can deduce the bounds that $\mathbb{P}(Y(1) = 1, Y(0) = 0|X = x) \leq \mathbb{P}(Y(1) = 1|X = x) + \mathbb{P}(Y(0) = 0|X = x)$ and $\mathbb{P}(Y(1) = 0, Y(0) = 1|X = x) \leq \mathbb{P}(Y(1) = 0|X = x) + \mathbb{P}(Y(0) = 1|X = x)$.

Applying the two equations in data $D$, we define the number of persuadable individuals in total $k$ samples as $N^P(k)$ and the number of sleeping dog individuals in total $k$ samples as $N^S(k)$, then we have the two bounds that $N^P(k) \leq R^T(D, k) + NR^C(D, k)$ and $N^S(k) \leq R^C(D, k) + NR^T(D, k)$.

The difference between the two bounds is the value function of principled uplift curve. In other words, the principle uplift measures the difference between the total number of samples that may be in the persuadable stratum and the total number of samples that may be in the sleeping dog stratum. Thus, the proof of Proposition 1.1 is done that the principled uplift becomes larger as the persuadable individuals increase, and becomes smaller as the sleeping dog individuals increase.

As demonstrated by the analysis in Tables 1 to 3, the principled uplift curve value associated with the number of treatment group samples that may be in the persuadable stratum equals the number of control group samples within the same stratum. Similarly, the value corresponding to the number of treatment group samples that could belong to the sleeping dog stratum equates to the value for the number of control group samples in the sleeping dog stratum. Consequently, the ranking of treatment and control groups does not affect the curve value of the principled uplift. The proof of proposition 1.2 is done.

## F. Principled Uplift Curve versus Conventional Curves

In this section, we transform the PUC into a form similar to the conventional curves to explore the differences between the two.

First, by substituting $N_S^T$ and $N_S^C$ into the PUC, we obtain:

$$\begin{aligned} V_{\text{PUC}}(k, S) &= R_S^T(D, k) + N_S^C(D, k) - R_S^C(D, k) - R_S^C(D, k) - N_S^T(D, k) + R_S^T(D, k) \\ &= N_S^C(D, k) - N_S^T(D, k) + 2(R_S^T(D, k) - R_S^C(D, k)). \end{aligned}$$

(29)

Based on this equation, the PUC can be interpreted as incorporating $N_S^T$ and $N_S^C$ to balance the number of individuals across different strata, with the key distinction being that it assigns weights differently compared to conventional curves.

We can also compare conventional metrics with the PUC metric from the perspective of the ranking rule $S_{\text{Max}}$.

For conventional curves, including SUC, SQC, JUC, and JQC, $S_{\text{Max}}$ is defined as:

$$S_{\text{Max}}(D_i) = \mathbb{I}(y_i = 1)\big(\mathbb{I}(t_i = 1) - \mathbb{I}(t_i = 0)\big),$$

(30)

whereas for PUC, the $S_{\text{Max}}$ is given by:

$$S_{\text{Max}}(D_i) = \mathbb{I}(y_i = 1)\big(\mathbb{I}(t_i = 1) - \mathbb{I}(t_i = 0)\big) + \mathbb{I}(y_i = 0)\big(\mathbb{I}(t_i = 0) - \mathbb{I}(t_i = 1)\big).$$

(31)

---

[6]The joint uplift curve is an exception, where dynamically changing numbers $N_S^T(D, k)$ and $N_S^C(D, k)$ may lead to catastrophic misrankings that intermix $TP$ and $CP$ subgroups.

It is evident that maximizing the PUC value requires a ranking rule that builds upon the conventional curve by prioritizing individuals with $T = 0, Y = 0$ and $T = 1, Y = 1$ at the top, while placing individuals with $T = 1, Y = 0$ and $T = 0, Y = 1$ at the bottom.

## G. The Variants of Principled Uplift Curve

The value function of the principled uplift curve in Equation (9) is formulated as follows:

$$
\begin{aligned}
V(k, S) = & R_S^T(D, k) + NR_S^C(D, k) \\
& - R_S^C(D, k) - NR_S^T(D, k).
\end{aligned}
\tag{32}
$$

The principled uplift curve (v1) is defined as follows:

$$
\begin{aligned}
V(k, S) = & \min\left(\frac{R_S^T(D, k)}{N_S^T(D, k)}, \frac{NR_S^C(D, k)}{N_S^C(D, k)}\right) \\
& - \min\left(\frac{NR_S^T(D, k)}{N_S^T(D, k)}, \frac{R_S^C(D, k)}{N_S^C(D, k)}\right).
\end{aligned}
\tag{33}
$$

The principled uplift curve (v2) is defined as follows:

$$
\begin{aligned}
V(k, S) = & \min\left(\frac{R_S^T(D, k)}{|T|}, \frac{NR_S^C(D, k)}{|C|}\right) \\
& - \min\left(\frac{NR_S^T(D, k)}{|T|}, \frac{R_S^C(D, k)}{|C|}\right),
\end{aligned}
\tag{34}
$$

where $|T|$ and $|C|$ represent the total number of individuals in the treatment group and the control group, respectively.

The principled uplift curve (v3) is defined as follows:

$$
\begin{aligned}
V(k, S) = & \max\left(\frac{R_S^T(D, k)}{|T|}, \frac{NR_S^C(D, k)}{|C|}\right) \\
& - \min\left(\frac{NR_S^T(D, k)}{|T|}, \frac{R_S^C(D, k)}{|C|}\right).
\end{aligned}
\tag{35}
$$

The principled uplift curve (v4) is defined as follows:

$$
\begin{aligned}
V(k, S) = & \max\left(\frac{R_S^T(D, k)}{|T|}, \frac{NR_S^C(D, k)}{|C|}\right) \\
& - \max\left(\frac{NR_S^T(D, k)}{|T|}, \frac{R_S^C(D, k)}{|C|}\right).
\end{aligned}
\tag{36}
$$

The principled uplift curve (v5) is defined as follows:

$$
\begin{aligned}
V(k, S) = & \min\left(\frac{R_S^T(D, k)}{|T|}, \frac{NR_S^C(D, k)}{|C|}\right) \\
& - \max\left(\frac{NR_S^T(D, k)}{|T|}, \frac{R_S^C(D, k)}{|C|}\right).
\end{aligned}
\tag{37}
$$

The principled uplift curve (v6) is defined as follows:

$$
\begin{aligned}
V(k, S) = & \left(\frac{R_S^T(D, k)}{|T|}, \frac{NR_S^C(D, k)}{|C|}\right) \\
& - \left(\frac{NR_S^T(D, k)}{|T|}, \frac{R_S^C(D, k)}{|C|}\right).
\end{aligned}
\tag{38}
$$

*Table 9.* The Kendall between AUUQC and AUTGC sequences of S-Learner on synthetic data. The symbol (↑) signifies that higher values are desirable. The SUC, SQC, JUC, JQC, and PUC denote their AUUQC values.

| $Kendall$ (↑) | S-Learner |
|---|---|
| SUC | 0.776 |
| SQC | 0.776 |
| JUC | 0.812 |
| JQC | 0.765 |
| PUC (v1) | 0.158 |
| PUC (v2) | 0.552 |
| PUC (v3) | 0.560 |
| PUC (v4) | 0.601 |
| PUC (v5) | 0.688 |
| PUC (v6) | 0.616 |
| PUC (v7) | 0.514 |
| PUC (v8) | 0.545 |
| PUC | **1.000** |

This version is similar to the ROCini in Verbeken et al. (2022).

The principled uplift curve (v7) is defined as follows:

$$V(k, S) = (\frac{R_S^T(D, k)}{N_S^T(D, k)}, \frac{NR_S^C(D, k)}{N_S^C(D, k)})$$
$$- (\frac{NR_S^T(D, k)}{N_S^T(D, k)}, \frac{R_S^C(D, k)}{N_S^C(D, k)}). \tag{39}$$

The principled uplift curve (v8) is defined as follows:

$$V(k, S) = (R_S^T(D, k) + NR_S^C(D, k))$$
$$- (NR_S^T(D, k) + R_S^C(D, k)). \tag{40}$$

Following Figure 5, we apply the Kendall coefficient between AUUQC and AUTGC to quantify the rank correlation between the sequences of these two metrics. As Table 9 shown, according to their performance, we select the value function of the principled uplift as $V(k, S) = R_S^T(D, k) + NR_S^C(D, k) - R_S^C(D, k) - NR_S^T(D, k)$.

## H. Analysis of PTONet

Next, we conduct a comprehensive analysis of PTONet from perspectives including computational complexity, scalability, potential overfitting issues, and detailed function of PTONet.

**Computational Complexity.** Based on the parameter tuning range we used, the number of parameters for PTONet ranges from 22,323 to 30,556,160, with floating point operations ranging from 1,578 to 29,090.

**Scalability.** Our study uses both a synthetic dataset and the real-world Criteo dataset, Lazada dataset to represent small and large-scale scenarios. As shown in Tables 4, 5 and 13, PTONet consistently outperforms other models on the PUC metric. As detailed in Table 10, the synthetic data, Criteo data, and Lazada data each present unique treatment and control ratios, positive outcome ratios, and varying numbers of covariates, allowing us to evaluate scalability.

**Potential Overfitting Issues.** The propensity estimator, treatment reconstructor, and outcome estimator modules of PTONet each help prevent the loss of CATE information due to overfitting. Additionally, we employ early stopping on the validation set to further prevent overfitting. As shown in Table 13, it can be observed that our method still outperforms other methods on the PUC metric even in high-dimensional real-world data.

**Detailed Function of PTONet.** The role of PTONet is to serve as a feasible model guided by the PUC metric. The novelty of its structure is not the focus of our paper, as its backbone is merely a variant of DragonNet (Shi et al., 2019). As shown in Table 4 of our paper, evaluates the S-Learner, S-Learner (PU), and PTONet, highlighting the performance impact of PUL and PTONet's backbone. Further detailed results are provided in Table 6. The PUL, treatment loss and balance loss notably boost the model's performance. As targeted regularizer loss is intended to improve model scalability on observational data, its lesser impact on RCT data is expected. Specifically, 'treatment assignment bias' refers to the potential bias that arises when the assignment of treatments is influenced by certain factors or variables, rather than being random. The most common methods to mitigate this bias include the use of propensity score matching, weighted regression, or doubly robust learning. The principle of the 'targeted regularizer' lies in its ability to eliminate treatment assignment bias through the introduction of propensity scores, while also ensuring doubly robust learning based on semi-parametric estimation theory. This module, not an original contribution of this paper, was first proposed in Shi et al. (2019).

We have not extensively discussed the concepts of treatment assignment bias and the targeted regularizer because, while the former is a common issue in causal inference, it is not the primary focus of this paper. We introduced these concepts mainly to ensure the scalability of our model, which can adapt whether or not treatment assignment bias exists.

## I. Experimental Details

**Dataset Description.** The synthetic dataset contains $n = 10,000$ units with $q = 10$ different covariates. For each unit $i$, the covariates are initially independently and identically sampled from a normal distribution $X_{ij} \sim \mathcal{N}(0,1)$ for $j \in \{1, 2, \cdots, q\}$. We generate the treatment assignment $T_i$ using $T_i \sim \text{Binomial}(1, 0.1)$, denoting the binomial distribution in a single trial with success probability $0.1$. It is designed to simulate the real-world scenario in our business data, where the number of treated samples is significantly smaller than the number of control samples. This also ensures differentiation from the Criteo and Lazada datasets.

We simulate potential outcomes based on the covariates as follows:

$$
Y_i(0) = 0.5 \sin(\sum_{j}^{q} X_i^j + 1) + \epsilon_i^0,
$$
$$
Y_i(1) = 0.1(\sum_{j}^{5} \cos(X_i^j) + 2) + \epsilon_i^1,
$$
(41)

where the noise terms $\epsilon_i^0, \epsilon_i^1 \sim \mathcal{N}(0, 0.1)$. Here, the sine and cosine functions are introduced to incorporate nonlinearity, while the different coefficients are used to adjust the proportion of samples with $\tau(x) > 0$. This adjustment helps simulate the real-world business scenario where positive outcomes are relatively rare.

The true ITE $\tau_i = Y_i(1) - Y_i(0)$ and CATE $\tau(x) = \mathbb{E}[Y_i(1) - Y_i(0)|X = x]$. The final observed outcome is generated as follows:

$$
Y_i = T_i \mathbb{I}(\tau_i > 0) + (1 - T_i)\mathbb{I}(\tau_i < 0) + \epsilon_i^y \mathbb{I}(\tau_i = 0),
$$
(42)

where $\epsilon_i^y \sim \text{Binomial}(1, 0.5)$. The first term represents the observed outcome for the persuadable group, the second term corresponds to the sleeping dogs, and the third term accounts for the observed outcome of sure things and lost causes. For details on the proportion of the treated group and the positive outcome rate, please refer to Table 10.

The real-world Criteo dataset (Diemert Eustache et al., 2018; Diemert et al., 2021), open sourced by Criteo AI Labs, is utilized for uplift modeling in a large-scale advertising scenario. It includes 25,309,483 instances with 11 continuous features, a binary treatment, and 2 candidate outcomes (visits and conversions).

The real-world Lazada data is large-scale production dataset from the real voucher distribution business scenario in Lazada, a leading South-East Asia (SEA) E-commerce platform of Alibaba Group. Our data processing follows Zhong et al. (2022).

The synthetic data, Criteo data, and Lazada data each present unique treatment and control ratios, positive outcome ratios, and varying numbers of covariates, allowing us to evaluate scalability.

**Hyperparameters Tuning.** The range of values for hyperparameters shared by all methods is presented as follows: the representation dimension $hdim \in \{2^4, 2^5, 2^6\}$, the batch size $bs \in \{2^8, 2^9, 2^{10}, 2^{11}\}$ and the learning rate $lr \in$

*Table 10.* Statistics of the synthetic dataset, Criteo dataset and Lazada dataset.

| Dataset | Covariates | Training Data | | | Testing Data | | |
|---|---|---|---|---|---|---|---|
| | | Treated (percentage) | Positive Outcome (percentage) | Total | Treated (percentage) | Positive Outcome (percentage) | Total |
| Synthetic | 10 | 910 (10.1%) | 3,913 (43.5%) | 9,000 | 92 (9.2%) | 447 (44.7%) | 1,000 |
| Criteo | 12 | 10.7M (85.0%) | 5.92K (4.70%) | 12.6M | 1.19M (85.0%) | 65.4K (4.68%) | 1.40M |
| Lazada | 83 | 0.92M (22.1%) | 83.0K (2.00%) | 4.17M | 0.47M (51.6%) | 31.9K (3.54%) | 0.91M |

$\{1e^{-4}, 1e^{-3}, 1e^{-2}, 1e^{-1}\}$. Furthermore, for the hyperparameters $\alpha$ in CFRNet, DragonNet, DESCN, and PTONet, $\beta$ in DragonNet and PTONet, and $\beta_0, \beta_1, \gamma_0, \gamma_1$ in DESCN, they are all confined to the range of $\{0.1, 0.5, 1, 5, 10\}$. We utilize an Adam optimizer with a maximum of 20 epochs and employ joint Qini as the primary evaluation metric. We implement an early stopping mechanism with patience of 5 for all baselines, as suggested by (Liu et al., 2023). Based on the parameter tuning range we used, the number of parameters for PTONet ranges from 22,323 to 30,556,160, with floating point operations ranging from 1,578 to 29,090.

**Implementation Details.** All experiments are conducted on an Intel(R) Xeon(R) Gold 6240 CPU @ 2.60GHz. We used the scikit-uplift package to conduct all AUUQC in this paper. The Kendall coefficient was computed using the 'kendalltau' function from the scipy.stats module.

**More Ablation Studies.** In addition to applying PUL directly to the S-Learner in Table 4, we further incorporate PUL into T-Learner, TARNet, and EUEN to more comprehensively validate its effectiveness. As Table 11 shown, The performance of these three models after incorporating the Principled Uplift loss function is comparable to that of S-Learner (PU) and PTONet, with significant improvement observed across all metrics. Furthermore, comparing the performance of the five models across various metrics reveals that, except for the PUC metric, the results of the other metrics are inconsistent with AUTGC, highlighting the advantage of PUC in aligning with ground truth. Notably, PTONet achieves the best performance on both the PUC and AUTGC metrics, further demonstrating its effectiveness.

*Table 11.* Ablation studies of different backbones on synthetic data. SUC, SQC, JUC, JQC, and PUC denote their AUUQC values. The symbol ($\downarrow$) indicates that lower values are preferable, while ($\uparrow$) signifies that higher values are desirable. "TRUE" represents the results of the ground truth CATE.

| Uplift Model | Unbalanced Conventional Evaluation Metric | | | | | Balanced Evaluation Metric | |
|---|---|---|---|---|---|---|---|
| | PEHE ($\downarrow$) | SUC ($\uparrow$) | SQC ($\uparrow$) | JUC ($\uparrow$) | JQC ($\uparrow$) | PUC ($\uparrow$) | AUTGC ($\uparrow$) |
| TRUE | 0.000 | 0.835 | 0.586 | 0.779 | 0.581 | 1.000 | 1.000 |
| S-Learner (PU) | 0.879 ± 0.16 | **0.786 ± 0.16** | **0.552 ± 0.11** | 0.738 ± 0.15 | **0.548 ± 0.11** | 0.943 ± 0.19 | 0.957 ± 0.15 |
| T-Learner (PU) | 0.867 ± 0.14 | 0.763 ± 0.17 | 0.536 ± 0.12 | 0.748 ± 0.15 | 0.537 ± 0.11 | 0.937 ± 0.20 | 0.952 ± 0.15 |
| TARNet (PU) | 0.893 ± 0.08 | 0.759 ± 0.12 | 0.533 ± 0.09 | **0.754 ± 0.11** | 0.534 ± 0.08 | 0.944 ± 0.14 | 0.957 ± 0.11 |
| EUEN (PU) | **0.781 ± 0.15** | 0.767 ± 0.16 | 0.538 ± 0.11 | 0.742 ± 0.15 | 0.538 ± 0.11 | 0.932 ± 0.19 | 0.948 ± 0.15 |
| PTONet | 0.883 ± 0.13 | 0.780 ± 0.14 | 0.547 ± 0.10 | 0.746 ± 0.13 | 0.546 ± 0.10 | **0.948 ± 0.15** | **0.961 ± 0.11** |

**More Case Studies.** To further explore potential issues in uplift modeling guided by either conventional curves or propensity score, we conduct an experiment comparing the performance of various models under same conditions, using a fixed random seed of zero. We observe several notable phenomena from the results presented in Table 12.

From an evaluation standpoint, firstly, when comparing the S-Learner (U) and the S-Learner (PS), we find that a low PEHE does not necessarily imply an accurate CATE ranking, highlighting the disparity between estimation and ranking issues in causal inference. Secondly, apart from PUC, the magnitudes of AUUQC for other curves do not align with AUTGC. Notably, in the case of S-Learner (U) and S-Learner (PS), the AUUQC values of SUC, SQC, and JQC are even larger than the the value of TRUE.

From a modeling perspective, S-Learner (PS) achieves higher AUUQC values of conventional curves than S-Learner. These phenomena suggest that models with higher AUUQC values of conventional curves may have learned biased treatment assignments. Conversely, S-Learner (PU) performs better than S-Learner (U) and S-Learner (PS) on the AUUQC of PUC and AUTGC, showing the feasibility of involving principled uplift to guide uplift modeling. Compared to Table 4, we find that although the best performance of the S-Learner (PU) is not better than the S-Learner (U) in the AUUQC of SUC, SQC,

*Table 12.* Performance comparison of different uplift models on synthetic data. SUC, SQC, JUC, JQC, and PUC denote their AUUQC values. The symbol (↓) indicates that lower values are preferable, while (↑) signifies that higher values are desirable. "TRUE" represents the results of the ground truth CATE. The symbol $^*$ indicates that the uplift model's evaluation result exceeds the ground truth result.

| Uplift Model | Unbalanced Conventional Evaluation Metric | | | | | Balanced Evaluation Metric | |
| --- | --- | --- | --- | --- | --- | --- | --- |
| | PEHE (↓) | SUC (↑) | SQC (↑) | JUC (↑) | JQC (↑) | PUC (↑) | AUTGC (↑) |
| TRUE | 0.000 | 0.835 | 0.586 | 0.779 | 0.581 | 1.000 | 1.000 |
| S-Learner | 1.392 | 0.297 | 0.208 | 0.332 | 0.210 | 0.371 | 0.519 |
| S-Learner (U) | 1.133 | **0.847**$^*$ | **0.594**$^*$ | 0.677 | 0.580 | 0.892 | 0.918 |
| S-Learner (PS) | 1.350 | 0.847$^*$ | 0.594$^*$ | **0.755** | **0.595**$^*$ | 0.915 | 0.935 |
| S-Learner (PU) | 1.158 | 0.783 | 0.549 | 0.752 | 0.547 | 0.946 | 0.959 |
| T-Learner | 1.254 | 0.596 | 0.418 | 0.631 | 0.426 | 0.680 | 0.755 |
| TARNet | 1.164 | 0.683 | 0.479 | 0.713 | 0.485 | 0.864 | 0.896 |
| CFRNet | 1.445 | -0.034 | -0.024 | 0.018 | 0.001 | 0.060 | 0.281 |
| DragonNet | 1.036 | 0.732 | 0.513 | 0.693 | 0.510 | 0.897 | 0.921 |
| EUEN | 1.177 | 0.602 | 0.422 | 0.526 | 0.410 | 0.688 | 0.761 |
| DESCN | 1.090 | 0.780 | 0.547 | 0.748 | 0.547 | 0.949 | 0.961 |
| EFIN | 1.751 | 0.619 | 0.434 | 0.599 | 0.431 | 0.744 | 0.804 |
| PTONet | **0.963** | 0.770 | 0.540 | 0.747 | 0.536 | **0.959** | **0.969** |

JUC, and JQC, the mean performance of S-Learner (PU) surpasses that of S-Learner (U). This indicates that models guided by the principled uplift curve yield more robust results.

**Results on Lazada Dataset.** We include results from the high-dimensional Lazada dataset (Zhong et al., 2022) in Table 13. Due to the small curve values in the dataset, we retain an additional decimal place when presenting the results for improved precision. We observe that although DESCN performs best on SUC, SQC, JUC, and JQC, PTONet consistently outperforms other models on the PUC metric in the Lazada dataset.

*Table 13.* Performance comparison (mean$_{\pm std}$) on Lazada dataset. PTONet achieves superior performance on PUC.

| Uplift Model | Unbalanced Conventional Evaluation Metric | | | | Balanced Metric |
| --- | --- | --- | --- | --- | --- |
| | SUC (↑) | SQC (↑) | JUC (↑) | JQC (↑) | PUC (↑) |
| S-Learner | $0.0875_{\pm 0.005}$ | $0.0256_{\pm 0.001}$ | $0.0035_{\pm 0.001}$ | $0.0247_{\pm 0.001}$ | $0.0052_{\pm 0.001}$ |
| T-Learner | $0.0684_{\pm 0.017}$ | $0.0200_{\pm 0.005}$ | $0.0031_{\pm 0.001}$ | $0.0219_{\pm 0.004}$ | $0.0066_{\pm 0.001}$ |
| TARNet | $0.0822_{\pm 0.015}$ | $0.0241_{\pm 0.004}$ | $0.0033_{\pm 0.001}$ | $0.0232_{\pm 0.005}$ | $0.0044_{\pm 0.001}$ |
| CFRNet | $0.0628_{\pm 0.029}$ | $0.0184_{\pm 0.009}$ | $0.0025_{\pm 0.001}$ | $0.0176_{\pm 0.008}$ | $0.0038_{\pm 0.002}$ |
| DragonNet | $0.0745_{\pm 0.003}$ | $0.0218_{\pm 0.001}$ | $0.0035_{\pm 0.001}$ | $0.0249_{\pm 0.002}$ | $\underline{0.0082_{\pm 0.001}}$ |
| EUEN | $\underline{0.0893_{\pm 0.012}}$ | $\underline{0.0262_{\pm 0.003}}$ | $\underline{0.0036_{\pm 0.001}}$ | $\underline{0.0253_{\pm 0.003}}$ | $0.0050_{\pm 0.001}$ |
| DESCN | $\mathbf{0.1160_{\pm 0.007}}$ | $\mathbf{0.0340_{\pm 0.002}}$ | $\mathbf{0.0041_{\pm 0.001}}$ | $\mathbf{0.0295_{\pm 0.002}}$ | $0.0007_{\pm 0.001}$ |
| EFIN | $0.0645_{\pm 0.013}$ | $0.0189_{\pm 0.004}$ | $0.0023_{\pm 0.001}$ | $0.0164_{\pm 0.002}$ | $0.0025_{\pm 0.003}$ |
| PTONet | $0.0424_{\pm 0.015}$ | $0.0124_{\pm 0.004}$ | $0.0025_{\pm 0.001}$ | $0.0174_{\pm 0.005}$ | $\mathbf{0.0108_{\pm 0.001}}$ |

## J. Limitations and Future Work

There are several directions for further exploration. Firstly, this paper only addresses the conventional setting of uplift modeling, precisely the scenario of binary treatment and binary outcome in RCT data. The performance of the principled uplift curve and PTONet in scenarios involving multi-valued or continuous treatment or outcome, and even in observational data settings, remains to be explored. Furthermore, the performance of the principled uplift curve relies on the assumption of

unconfoundedness. The extent of the presence of unobserved confounders remains a crucial question. Finally, the principled uplift curve in this paper only achieves the division of the persuadable and sleeping dog groups without the capability to identify sure things and lost causes. The presence of sure things and lost causes can also affect the performance of the principled uplift curve. Although one way to mitigate this effect is to filter out samples with estimated CATEs of 0 using an uplift model beforehand and then evaluate the remaining samples uniformly, we still anticipate the emergence of a new metric in the future that can address this issue more comprehensively.

