# OpenReview forum: "Rethinking Causal Ranking: A Balanced Perspective on Uplift Model Evaluation"
_ICML.cc/2025/Conference — ICML 2025 poster_

### Official Review · Reviewer_MgqQ · 2025-03-12

**Overall Recommendation:** 4

**Summary:**

This work focuses on building and evaluating models for uplift modeling. The work finds a critical limitation in existing evaluation metrics, as many of these models do not weigh negative outcomes enough. The work finds that this lead to biased evaluations due to incorrect orderings between persuadable and sleeping dogs with negative outcomes, potentially resulting in biased models receiving higher curve values. The authors show this through both empirical results and theoretical results. Given the limitation of existing evaluation metrics, the work proposes the principled uplift curve (PUC), and show that it properly weighs different individuals in both the positive and negative outcome groups. The authors propose PTONet by integrating the PUC into the objective function to reduce bias during uplift model training. Through experimental results, the efficacy of PUC is shown by its alignment with ground-truth evaluation in synthetic settings, and the efficacy of PTONet is established in both synthetic and real data.

## Update after Rebuttal

I appreciate the authors work in answering the questions. Overall, I am satisfied with the response and have updated my score. I do hope the authors consider the limitations brought up by other reviewers and discuss them to ensure the paper does not over claim the contributions of the proposed metric (which I do not believe they currently do).

**Claims And Evidence:**

The primary claims of this work are:

1) Existing uplift model evaluation curves can result in suboptimal ordering
2) The proposed PUC provides a more balanced and unbiased evaluation compared to regular uplift and Qini curves
3) The proposed PTONet enhances a model's ability to rank CATEs effectively.

The authors provide sufficient evidence for all of these claims.

**Essential References Not Discussed:**

I would appreciate more discussion on [1]. In the evaluation metrics considered in [1], every example will contribute to the TOC/AUTOC curve, which may mitigate some of found issues with most uplift modeling metrics.

[1] Yadlowsky, Steve, et al. "Evaluating treatment prioritization rules via rank-weighted average treatment effects." Journal of the American Statistical Association (2024): 1-14.

**Experimental Designs Or Analyses:**

The experimental design and analyses seem sound overall.

**Methods And Evaluation Criteria:**

The proposed methods and evaluation criteria make sense for the problem at hand, and the inclusion of both synthetic data and real data is appreciated.

**Other Comments Or Suggestions:**

N/A

**Other Strengths And Weaknesses:**

Strengths:

1. The authors find a really subtle yet interesting flaw in how most uplift evaluation metrics are evaluated
2. The clear examples of when an unbiased model is rated worse than a biased model in Table 2/3 is very compelling
3. The correlation between the ground-truth AUTGC and PUC in the experimental results is very convincing, and seems to support the theoretical findings for PUC
4. PTONet has strong performance in terms of PUC. The fact that the next performing model is PU S-Learner shows more credence to the authors findings regarding how to optimize the PUC.
5. The ablations of the proposed method are convincing in terms of showing the importance of every part of the objective

Weaknesses and Concerns:

1. The work does not mention the relationship to [1] (i.e., does TOC overcome many of the issues of past work?)
2. The organization of the proposed method is quite difficult to follow. Specifically, it is not clear how the loss in (11) is formulated. A more clear description of this would be useful as this is the crux of the proposed method.
3. The proof of Proposition 4.1 is not well formulated and very difficult to follow. Once again, as this is a major part of the proposed contribution, a more clear proof for this proposition would be appreciated.

**Questions For Authors:**

1. When proving Proposition 4.1, how does the difference between the two bounds prove the first half of Proposition 4.1? I would appreciate seeing the steps more clearly.
2. What is the exact intuition for why the loss (11) is useful? What does identifying g(t_i, y_i) from the estimated treatment effect help guide models to assign higher CATEs to persuadable individuals?

**Relation To Broader Scientific Literature:**

The ability to accurately evaluate uplift models is critical across many fields, including marketing and advertising where personalized decision-making is key. The contributions in finding the weaknesses of existing evaluation metrics as well as the new proposed evaluation metric are hence quite important. The proposed PTONet also shows a new way to improve uplift modeling, which may be used in these downstream applications as well.

**Theoretical Claims:**

I examined the correctness of proposition 4.1. Truthfully, it is a bit difficult to follow. The rough steps make sense, though it is not clear why we define the value function has the difference between the two bounds -- this is stated without justification in the proof. The proof for the second half of 4.1 is more clear.

---

> ### Author Rebuttal · Authors · 2025-03-29
>
> **References &Weaknesses 1:** I would appreciate more discussion on [1].
>
> **Response 1:** Thank you for your concern. As far as we know, TOC/AUTOC can be understood as introducing a threshold $u$ and a logarithmic function to the conventional uplift and Qini curves, as shown in the following formula (from Equation (2.5) in [1]):
>
> $\begin{aligned} & \operatorname{AUTOC}(S)=\mathbb{E}\left[\left(-\log \left(1-F_S\left(S\left(X_i\right)\right)\right)-1\right)\left(Y_i(1)-Y_i(0)\right)\right] \\ & \operatorname{QINI}(S)=\mathbb{E}\left[\left(F_S\left(S\left(X_i\right)\right)-\frac{1}{2}\right)\left(Y_i(1)-Y_i(0)\right)\right]\end{aligned}$
>
> This indicates that the metric **places particular emphasis on the contribution of the top few individuals** (as shown in Figure 2 of [1], where, if only the top 10% of the population is considered, the overall gain from AUTOC is higher than that from QINI). In other words, this metric **amplifies the imbalance issue inherent in the uplift and Qini curves.** In contrast, the goal of our metric is the opposite—**we aim to address and mitigate this imbalance problem.**
>
> **Weaknesses 2&Questions 2:** What is the exact intuition for why the loss (11) is useful? What does identifying g(t_i, y_i) from the estimated treatment effect help guide models to assign higher CATEs to persuadable individuals?
>
> **Response 2:** Thank you for your question. The intuition behind the loss function in equation (11) is derived from Table 1. Specifically, **the TP ($T=1, Y=1$) and CN ($T=0, Y=0$) samples should be ranked ahead of TN ($T=1, Y=0$) and CP ($T=0, Y=1$).**
>
> To incorporate this constraint during training, we introduce $g(t_i, y_i)$ as a binary classification task label, where the labels for TP and CN samples are 1, and the labels for TN and CP samples are 0. Then, we use the estimated causal effect $\hat{\tau}(x)$ as the input to train this binary classification task. This approach effectively constrains the model such that **the estimated causal effects $\hat{\tau}(x)$ for TP and CN are as large as possible, while for TN and CP, $\hat{\tau}(x)$ should be as small as possible.** In this way, when the trained model is tested or during model selection, the model will rank TP and CN ahead of TN and CP based on $\hat{\tau}(x)$ in descending order. The persuadable group corresponds to the TP and CN samples, while the sleeping dog group corresponds to the TN and CP samples.
>
> **Claims&Weaknesses 3&Questions 1:** The proof of Proposition 4.1 is not well formulated and very difficult to follow. When proving Proposition 4.1, how does the difference between the two bounds prove the first half of Proposition 4.1?
>
> **Response 3:** Thank you for your concern. Below, we will focus on explaining the meaning of these two bounds and why the difference between them is able to distinguish between the persuadable group and the sleeping dog group.
>
> In appendix E, we define the number of persuadable individuals in total $k$ samples as $N^P(k)$ and the number of sleeping dog individuals in total $k$ samples as $N^S(k)$, then we have the two bounds that $N^P(k) \le R^T(D,k) + NR^C(D,k)$ and $N^S(k) \le R^C(D,k) + NR^T(D,k)$.
>
> The first bound holds because the **persuadable group $(\tau(x) > 0)$ only includes the TP ($T=1, Y=1$) and CN ($T=0, Y=0$) groups.** Therefore, the number of individuals in the persuadable group, $N^P(k)$, will always be less than or equal to the sum of the number of TP individuals, $R^T(D,k)$, and CN individuals, $NR^C(D,k)$. **Similarly, the sleeping dog group only includes the TN ($T=1, Y=0$) and CP ($T=0, Y=1$) groups.** Thus, $N^S(k) \le R^C(D,k) + NR^T(D,k)$.
>
> We aim for the evaluation metric PUC to correctly distinguish between the persuadable group and the sleeping dog group. This means we want **PUC to increase as the persuadable group grows, and decrease as the sleeping dog group increases.** Therefore, we define:
>
> $V_{\operatorname{PUC}}(k,S) = R^T_{S}(D,k) + NR^C_{S}(D,k) - R^C_{S}(D,k) - NR^T_{S}(D,k).$
>
> As we can see, the first two terms include all the individuals in the persuadable group, while the last two terms include all individuals in the sleeping dog group. **When the persuadable group is included in PUC, the value of PUC increases**; conversely, **when the sleeping dog group is included in PUC, the value of PUC decreases**. Thus, our metric can effectively distinguish between the persuadable group and the sleeping dog group.
>
> We will include this explanation in Appendix E of the final version of the paper.
>
> Thank you once again for your valuable feedback. If you have any further concerns or questions, we are always happy to address them. If you feel that our responses have addressed your concerns, we would appreciate it if you could consider raising your recommendation score.

---

### Official Review · Reviewer_JDoP · 2025-03-13

**Overall Recommendation:** 3

**Summary:**

This paper proposes PTONet, a new uplift model that integrates the Principled Uplift Loss (PUL) to improve CATE ranking accuracy, outperforming existing models in experiments on simulated and real-world datasets.

**Claims And Evidence:**

The paper effectively presents its claims and supports them with clear evidence.

**Essential References Not Discussed:**

None

**Experimental Designs Or Analyses:**

The paper's experimental design appears methodologically sound.

**Methods And Evaluation Criteria:**

Yes, the paper proposes a method to improve CATE ranking accuracy.

**Other Comments Or Suggestions:**

None

**Other Strengths And Weaknesses:**

None

**Questions For Authors:**

Please refer to the above.

**Relation To Broader Scientific Literature:**

The key contributions of this paper are positioned within the broader context of uplift modeling, which has been a subject of significant interest in domains like marketing, customer retention, and personalized treatment recommendations.

**Theoretical Claims:**

The theoretical claims and their proof are correct.

---

> ### Author Rebuttal · Authors · 2025-03-29
>
> Thank you for your feedback. If you have any additional concerns or questions, we would be happy to answer them. If you have no additional concerns, we would appreciate you considering increasing your recommendation score.

---

> > ### Comment · Reviewer_JDoP · 2025-04-04
> >
> > I confirm that I have read the author's response to my review and will update my review in light of this response as necessary.

---

### Official Review · Reviewer_e4Zy · 2025-03-16

**Overall Recommendation:** 3

**Summary:**

In an RCT with two groups, treatment and control, an uplift model is supposed to rank four types of units - treatment positive, treatment negative, control positive and control negative in alignment with their CATE (which is unobservable). This paper claims that existing evaluation metrics such as uplift curves and Qini curves are biased towards treating all negatives the same way, i.e., they ignore the potential that a control negative could potentially be a treatment positive and must be ranked at par with treatment positives. It then proposes a simple fix to the evalution metrics that leads to the proposed 'Principled Uplift Curve'. This idea is then also used to add an additional loss function for uplift modeling. Experimental results show that the proposed fix correlates best with true CATE rankings.

**Claims And Evidence:**

I don't see enough clear explanation for the claim that equation 12 handles treatment assignment bias. While this is not the main point of the paper, given that it is included and important for the PTONet (the proposed uplift model), either it should be explained more or if it is from previous work, clear citations should be added.

**Essential References Not Discussed:**

None that I know of.

**Experimental Designs Or Analyses:**

The experimental design is sound as per my understanding. A few issues a) I couldn't understand why the outcome in the synthetic data, in Appendix I is real-valued when the rest of the analysis is for a binary outcome. b) Also, the form of the functions don't seem to have a rationale mentioned in the text.

**Methods And Evaluation Criteria:**

The paper compares its fix against multiple uplift models and evaluation criteria present in literature. A few specific issues a) Why does the architecture in Figure 4 have the input from h(X,T) being added to g(T,Y). It seems like in equation 11, the BCE only takes in $\sigma(x)$. b) the paper needs to address the text on treatment bias with some more explanation.

**Other Comments Or Suggestions:**

1) Section 2.2 has some notation that is not right. I(k) is assumed to be an ordered index but it is not made clear what the order is when the index i ranges from 1 to I(k). Is I_diff same as I?
2) SUC in line 124 occurs before it is defined below.
3) The plot in Figure 5 needs some more explanation about what the shaded region is and what the lines are.

**Other Strengths And Weaknesses:**

Overall, the main contribution is to propose a fix to existing evaluation metrics in the uplift modeling literature. This is certainly important and impactful. The paper is written clearly overall. There are some parts which need more work that I specify later. My overall impression was that the proposed fix was 'obvious' and what anyone should do in the first place. On one hand, the presentation of the problem such that the solution is obvious, is a strength of the paper. However, in this case, I find that the paper lacks any further insight apart from this fix.

**Questions For Authors:**

I have mentioned most questions in the previous sections. Regarding the results, even the S-Learner with the loss function fix seems to be competitive with PTONet for the fixed evaluation metrics. It would interesting to see if this holds for other uplift models too.

###update after rebuttal ###

I am satisfied with the responses that the authors provided. On the one hand, the experiments that they performed underscore an important point that existing learners with the loss function fix seem to be competitive with their proposed learner. However, on the other hand I feel it takes away from the utility of PTONet. So, I'll keep my score unchanged.

**Relation To Broader Scientific Literature:**

This work challenges existing evaluation metrics in the uplift modeling literature. It draws specifically from Devriendt et. al. 2020 that uses helper functions to come up with loss functions for training uplift models.

**Theoretical Claims:**

The main theoretical claim that the paper makes is that the proposed uplift curve is sound in its ranking. This follows immediately from the proposed fix.

---

> ### Author Rebuttal · Authors · 2025-03-29
>
> Thank you for your positive feedback. We will address each of your concerns one by one.
>
> **Claims&Methods:** ... explanation for the claim that equation 12 handles treatment assignment bias. ... clear citations should be added.
>
> **Response 1:** Thank you for your suggestion. We will revise the citation in line 315 of the original text to **"(please refer to Section 3 in Shi et al. (2019))"** for better readability.
>
> Treatment Assignment Bias occurs when treatment assignment is influenced by systematic factors rather than being entirely random, potentially leading to biased results. Since this paper focuses on RCT data, where treatment is assumed to be random, Treatment Assignment Bias is not a concern. The **Targeted Regularizer** in PTONet improves scalability to non-RCT data, enhancing its applicability in industry and future research.
>
> **Methods:**  a) Why does the architecture in Figure 4 have the input from h(X,T) being added to g(T,Y).
>
> **Response 2:** Thank you for your question. The function $\sigma(x)$ can be derived as:
>
> $\sigma(x) = \frac{1}{1+\exp(-\hat\tau(x))} = \frac{1}{1+\exp(-(h_Y(x,1)-h_Y(x,0)))},$
>
> The function $h_Y$ corresponds to the arrow in Figure 4. To avoid this ambiguity, we will modify the arrow in Figure 4 from **$h(X,T)\rightarrow g(T,X)$** to **$h_Y(X,T)\rightarrow g(T,X)$**.
>
> **Analysis:** a)  why the outcome in the synthetic data is real-valued. b)  the form of the functions don't seem to have a rationale.
>
> **Response 3:** Apologies for any confusion in your reading. We forgot to emphasize in Appendix I that the final observed outcome is generated as follows:
>
> $Y_i = T_i \mathbb{I}(\tau_i > 0) + (1 - T_i) \mathbb{I}(\tau_i < 0) + \epsilon_i^y \mathbb{I}(\tau_i = 0) $
>
> where $\epsilon_i^y \sim \operatorname{Binomial}(1, 0.5)$. The first term represents the observed outcome for the **persuadable** group, the second term corresponds to the **sleeping dogs**, and the third term accounts for the observed outcome of **sure things** and **lost causes**.
>
> The rationale behind this data generation process is as follows:
>
> $T_i$ is designed to simulate the real-world scenario in our business data, where **the number of treated samples is significantly smaller** than the number of control samples.
>
> In outcome functions, the sine and cosine functions are introduced to incorporate nonlinearity, while the different coefficients are used to adjust the proportion of samples with $\tau(x) > 0$. This adjustment helps simulate our real-world business scenario where **positive outcomes are relatively rare.**
>
> We will include these details in Appendix I of the final version of the paper.
>
>  **Comments 1:** what the order is when the index i ranges from 1 to I(k).
>
> **Response 4:** Thank you very much for your comments. **The ordered index used in $I(k)$ is based on the descending order of the score function $S$.** Due to the character limit of the response, please refer to Appendix C of the paper, where we provide a detailed explanation of the calculation process for $I(k)$.
>
> **Comments 1&2:** Is I_diff same as I? SUC in line 124 occurs before it is defined below.
>
> **Response 5:** Thank you for your correction. The term $I_{diff}$ is a typo and will be revised to $I$ in the final version. Similarly, "SUC" in line 124 will be corrected to "uplift and Qini curve."
>
> **Questions:**  It would interesting to see if this holds for other uplift models too.
>
> **Response 7:** Thank you for your question. We have additionally included experiments with **T-Learner, TARNet, and EUEN models**. The results are as follows:
> | Synthetic      | PEHE (↓)     | SUC (↑) | SQC (↑) | JUC (↑) | JQC (↑) | PUC (↑) | AUTGC (↑) |
> | -------------- | ------------ | ------------------- | ----------------- | ---------------- | -------------- | --------------------- | ------------ |
> | T-Learner (PU) | 0.867 ± 0.14 | 0.763 ± 0.17        | 0.536 ± 0.12      | 0.748 ± 0.15     | 0.537 ± 0.11   | 0.937 ± 0.20          | 0.952 ± 0.15 |
> | TARNet (PU)    | 0.893 ± 0.08 | 0.759 ± 0.12        | 0.533 ± 0.09      | 0.754 ± 0.11     | 0.534 ± 0.08   | 0.944 ± 0.14          | 0.957 ± 0.11 |
> | EUEN (PU)      | 0.781 ± 0.15 | 0.767 ± 0.16        | 0.538 ± 0.11      | 0.742 ± 0.15     | 0.538 ± 0.11   | 0.932 ± 0.19          | 0.948 ± 0.15 |
> | PTONet         | 0.883 ± 0.13 | 0.780 ± 0.14        | 0.547 ± 0.10      | 0.746 ± 0.13     | 0.546 ± 0.10   | 0.948 ± 0.15          | 0.961 ± 0.11 |
>
> The performance of these three **models after incorporating the Principled Uplift loss function is comparable to that of PTONet**, with significant improvement observed across all metrics. We will add these experiments in the final version of this paper.
>
> Thank you once again for your valuable feedback. If you have any further concerns or questions, we are always happy to address them. If you feel that our responses have addressed your concerns, we would appreciate it if you could consider raising your recommendation score.

---

### Official Review · Reviewer_gjPZ · 2025-03-18

**Overall Recommendation:** 2

**Summary:**

This paper reveals the limitations of previous uplift and Qini curves in evaluating uplift models, demonstrating their susceptibility to manipulation by suboptimal ranking strategies that can artificially enhance the performance of biased models. To address this, the authors introduce the Principled Uplift Curve (PUC), a metric that accounts for both positive and negative outcomes, offering a new assessment of uplift models. Additionally, they propose PTONet, a PUC-guided uplift model that optimizes uplift predictions by directly maximizing the PUC value.

**Claims And Evidence:**

Yes, the claims made in the submission are supported by clear and convincing evidence.

**Essential References Not Discussed:**

No.

**Experimental Designs Or Analyses:**

The paper evaluates the performance of uplift models and their corresponding evaluation metrics using both synthetic and real-world datasets. It conducts experiments on a synthetic dataset and the Criteo dataset (Diemert Eustache et al., 2018; Diemert et al., 2021) to assess model effectiveness in practical scenarios. Additionally, to examine the scalability of the proposed method in high-dimensional settings, the paper presents further experimental results on the Lazada dataset (Zhong et al., 2022).

**Methods And Evaluation Criteria:**

Yes, the proposed methods and evaluation criteria make sense for the problem or application at hand.

**Other Comments Or Suggestions:**

No.

**Other Strengths And Weaknesses:**

The paper should provide a stronger justification for the proposed methods and explain why they outperform existing metrics. A key concern is that the approach does not improve the worst-case scenario, where ST^TP and ST^CP are ranked first. In many real-world applications, such as advertising and recommendation systems with budget constraints, this ranking can lead to significant opportunity costs. In contrast, conventional metrics at least ensure that PE^{TP} is ranked no lower than second place. While the proposed method distinguishes persuadable individuals from sleeping dogs, it also introduces a tradeoff in opportunity cost and does not guarantee improved decision-making performance in practical applications.

It is not surprising that the proposed PTONet achieves the highest PUC and AUTGC, as it is specifically designed based on PUC. However, it does not outperform other methods on alternative evaluation metrics.

**Questions For Authors:**

Please see the weaknesses.

**Relation To Broader Scientific Literature:**

The paper contributes to the broader literature on uplift modeling methods and evaluation metrics for uplift models. It examines limitations in existing evaluation approaches, particularly the susceptibility of uplift and Qini curves to biased rankings. By introducing the Principled Uplift Curve (PUC) and the PTONet model, the paper offers a refined evaluation metric and an optimization-based modeling approach, adding to the ongoing research on uplift modeling and causal inference.

**Theoretical Claims:**

I did not thoroughly check each proof. However, they do not contradict my prior understanding.

---

> ### Author Rebuttal · Authors · 2025-03-29
>
> Thank you for your feedback. We will address each of your concerns one by one.
>
> **Weak 1:** A key concern is that the approach does not improve the worst-case scenario, ... conventional metrics at least ensure that PE^{TP} is ranked no lower than second place. ..., it also introduces a tradeoff in opportunity cost and does not guarantee improved decision-making performance in practical applications.
>
> **Response 1:** Thank you for your concern. As stated in the original manuscript, although PUC is not a perfect metric—it cannot fully identify the four groups—**it is more suitable than conventional metrics for evaluating uplift models.**  When the budget is limited, in the worst-case scenario, the PUC metric **at least ensures less harmful decisions** compared to conventional metrics. In the best-case scenario, PUC guarantees **both less harmful decisions and the highest possible gains**. Therefore, our metric outperforms conventional metrics.
>
> Specifically, regarding 'conventional metrics ensuring that $PE^{TP}$ is ranked no lower than second place', it seems to overlook **the severe issue where $PE^{CN}$ is ranked behind $SD^{TN}$.** This means that using regular metrics for causal ranking in an uplift model forces decision-makers to target all potential customers ($PE^{TP}$ and $PE^{CN}$) at the expense of customers who would have originally clicked or purchased ($SD^{TN}$). Such a strategy should be avoided in practice as **it harms a large group of customers.**
>
> Even with a budget that covers only $PE^{TP}$ and not $SD^{TN}$, the regular metrics-guided model **loses a large number of potential customers, $PE^{CN}$**. This misstep can mislead decision-makers into believing there are no further growth opportunities when, in fact, potential customers remain untapped. (Please refer to Figure 2 in the original paper; in the worst-case scenario, samples from **3. $SD^{TN}$ to 6. $PE^{CN}$** are considered neither beneficial nor harmful under conventional metrics.)
>
> In contrast, the PUC metric does not exhibit this issue **in the worst-ranking case**. If decision-makers realize that a small-budget promotion won't yield incremental returns, they can continue to expand the budget until the PUC slows down or the promotion is scaled back, **without harming customer interests or missing potential customers**. (Refer to Figure 3 in the original paper; decision-makers can clearly identify that only the groups from **1. $ST^{TP}$** to **4. $PE^{CN}$** are yielding benefits.)
>
> **Most importantly, in the best ranking case, PUC guided models can achieve the highest-gain decisions with the minimal budget, while conventional metrics cannot.**
>
> Therefore, the **PUC metric should be used** over regular metrics to select an uplift model that accurately targets potential customers without alienating existing customers who are willing to purchase. **Future work can improve upon the limitation of PUC's inability to identify the four groups, but regular curves should no longer be used for uplift model evaluation.**
>
> Finally, we appreciate you for suggesting the application scenarios in advertising and recommendation.
>
> **Weak 2:** It is not surprising that the proposed PTONet achieves the highest PUC and AUTGC, as it is specifically designed based on PUC. However, it does not outperform other methods on alternative evaluation metrics.
>
> **Reponse 2:**  Thank you for your concern. Our simulated data is not specifically designed for PUC; Our data generation process is simple and easy to follow:
>
> $T_i \sim \operatorname{Binomial}(1,0.1)$ is designed to simulate the real-world scenario in our business data, where **the number of treated samples is significantly smaller** than the number of control samples. This also ensures differentiation from the Criteo and Lazada datasets.
>
> The outcome functions are defined as:
>
> $Y_i(0) = 0.5\sin\left(\sum_j^q X_i^j + 1\right) + \epsilon^0_i $
>
> $Y_i(1) = 0.1\left(\sum_j^5 \cos(X_i^j) + 2\right) + \epsilon^1_i $
>
> Here, the sine and cosine functions are introduced to incorporate nonlinearity, while the different coefficients are used to adjust the proportion of samples with $\tau(x) > 0$. This adjustment helps simulate our real-world business scenario where **positive outcomes are relatively rare.**  For details on the proportion of the treated group and the positive outcome rate, please refer to Table 10 in the original paper.
>
> PTONet performs suboptimally with regular metrics but achieves the best performance with the PUC metric, which further **validates the issue of bias in regular metrics highlighted in this paper, and confirms that the PU loss function can directly improve the model's performance on the PUC metric.**
>
> Thank you once again for your valuable feedback. If you have any further concerns or questions, we are always happy to address them. If you feel that our responses have addressed your concerns, we would appreciate it if you could consider raising your recommendation score.

---

### Official Review · Reviewer_AASw · 2025-03-23

**Overall Recommendation:** 3

**Summary:**

This paper proposes a new evaluation metric, the Principled Uplift Curve (PUC), which assigns equal importance to individuals with positive and negative outcomes and offers an unbiased evaluation of uplift models. The authors derive a new loss function with a new model architecture to reduce bias during
uplift model training.

**Claims And Evidence:**

The paper claims that the traditional uplift and Qini curves might lead to biased evaluations and proposes the Principled Uplift Curve (PUC) and compares it with other evaluation metrics by their correlation with AUTGC.

The proposed method is demonstrated to outperform the existing method with both synthetic and real-world datasets.

**Essential References Not Discussed:**

I'm not aware of any essential references that are not discussed.

**Experimental Designs Or Analyses:**

I checked the experiment for the proposed model and evaluation metrics.

The model is shown to outperform the existing models in the proposed evaluation metric.

The proposed metric is shown to be more reliable with synthetic data. I think there could be more discussion and careful experiments on this.

**Methods And Evaluation Criteria:**

The proposed architecture is evaluated in multiple real-world datasets benchmarks and evaluation metrics.

**Other Comments Or Suggestions:**

NA

**Other Strengths And Weaknesses:**

Pros:

- The paper tackles a challenging problem in uplift modeling and proposes a method to improve the modeling and evaluation.
- The proposed method is evaluated with many real-world benchmarks

Cons:
The proposed evaluation metric could be discussed in more detail since it is an essential part of the proposed method.

**Questions For Authors:**

- Could you give some intuition about the discussion in section 3? What does the max curve indicate here in Fig 3.

- Is the experiment for the correlation between AUUQC and AUTGC as described in Appendix I? Do the results still hold for other data distribution?

**Relation To Broader Scientific Literature:**

Evaluation is a challenging problem in this area. This paper proposes a new evaluation metric and shows it is more reliable than the existing ones, which could be a nice contribution to uplift modeling.

**Theoretical Claims:**

I checked the derivation of individual contributions in Appendix D and did not find any issue with it.

---

> ### Author Rebuttal · Authors · 2025-03-29
>
> Thank you for your positive feedback. We will address your concerns and questions one by one.
>
> **Cons:** The proposed evaluation metric could be discussed in more detail since it is an essential part of the proposed method.
>
> **Response 1:** Thank you for your concern. Based on your suggestion, we provide additional discussion on the evaluation metric as follows:
>
> - **Providing intuition behind** $S_{Max}(D_i)$:
>   $S_{Max}(D_i)$ represents the maximum PUC value, which is achieved when all samples with $y = 1$ and $t = 1$, as well as those with $y = 0$ and $t = 0$, are ranked ahead of samples with $y = 1$ and $t = 0$, as well as those with $y = 0$ and $t = 1$.
>
> - **Repositioning the intuition behind the PUC metric (Proposition 4.1):**
>   We will move the first paragraph of explanation after Proposition 4.1 to the paragraph after formula (9) to improve readability.
>
> - **Clarifying the intuition behind** $g(t_i, y_i)$:
> We will clarify that $ g(t_i, y_i) $ assigns a value of 1 to samples with $ y = 1 $ and $ t = 1 $, as well as those with $ y = 0 $ and $t = 0$, while samples with $y = 1$ and $t = 0$, as well as those with $y = 0$ and $t = 1$, are assigned a value of 0.
> Using $g(t_i, y_i)$ as the label, we train a binary classifier to constrain $\hat{\tau}(x)$, ensuring that samples with $y = 1$ and $t = 1$, as well as those with $y = 0$ and $t = 0$, have a larger $\hat{\tau}(x)$, whereas samples with $y = 1$ and $t = 0$, as well as those with $y = 0$ and $t = 1$, have a smaller $\hat{\tau}(x)$.
>
> - **Providing intuition behind** $L^{PU}(D)$:
>   This loss function encourages the CATE of samples with $y = 1$ and $t = 1$, as well as those with $y = 0$ and $t = 0$, to be greater than the CATE of samples with $y = 1$ and $t = 0$, as well as those with $y = 0$ and $t = 1$.
>
> We appreciate your feedback and will incorporate these intuitions in the final version of our paper.
>
> **Question 1:** Could you give some intuition about the discussion in section 3? What does the max curve indicate here in Fig 3.
>
> **Response 2:**  Thank you for your question. The **intuition** behind the discussion in Section 3 is that we aimed to verify **whether SUC and other regular metrics reach their maximum values only when the causal effect ranking is completely accurate**. If this were the case, then SUC would be a reliable metric. However, we found that **even when the causal effect ranking is entirely correct, SUC does not always attain its maximum value.** On the contrary, certain **incorrect causal effect rankings can lead to SUC achieving its highest score** (refer to Tables 2 and 3). This observation led us to further investigate SUC and related formulas, ultimately inspiring this paper.
>
> In Figure 3, the max curve corresponds to $S_{Max}(D_i)$ in Equation (8). We have supplemented its underlying intuition in Response 1.
>
> **Question 2:** Is the experiment for the correlation between AUUQC and AUTGC as described in Appendix I? Do the results still hold for other data distribution?
>
> **Response 3:** Thank you for your question. Yes, the experimental setup in this paper is described in Appendix I. **The results still hold for other data distributions, as long as the data is RCT data.**
>
> Thank you once again for your valuable feedback. If you have any further concerns or questions, we are always happy to address them. If you feel that our responses have addressed your concerns, we would appreciate it if you could consider raising your recommendation score.

---

### Decision · Program_Chairs · 2025-05-01

**Decision:**

Accept (poster)

**Comment:**

The paper discusses the limitations of Qini curves/uplift models, focusing on bias in evaluation and offering a solution through PUC. Reviewers agreed that evaluation is challenging in this area. Authors adequately addressed concerns that were raised during the discussion including that the approach does not improve the worst-case scenario, and relation to Yadlowsky et al AUTOC evaluation measure.